Methods

# Intravital imaging of the formation and resolution of MHC class II–positive T-cell activation niches

David Oleksyn, Jim Miller

Intravital imaging has revealed many of the cellular interactions that regulate immune responses, but is limited by the number of cells that can be simultaneously identified and often restricted to analysis of a single time point. We have developed two new fluorescent reporter strains, IEbeta-mAmetrine and CD8beta-LSSmOrange, that faithfully label cells expressing MHC class II and CD8-positive conventional T cells, respectively. These fluorescent proteins are spectrally distinct from commonly used fluorescent proteins (GFP, YFP), and so, these mice can be used in combination with many previously created reporter mice. In addition, we established a protocol where we can sequentially image the same area of the ear dermis over several weeks without inducing additional inflammation. We applied these techniques to IEbeta-mAmetrine mouse co-expressing markers for CD11c, CXCL10, and CD4 T cells to quantify the formation of CXCL10-positive cell clusters, elaboration of different MHC class II–positive cells within these clusters, accumulation of CD4 T cells within these clusters, and the dissipation of these T-cell activation niches as the inflammatory response wanes.

## Introduction

An effective immune response requires sequential interactions between different populations of migratory immune cells within lymphoid tissue and again at peripheral sites of inflammation and infection. Recent imaging experiments, in particular intravital imaging (IVM), have shown that many of these interactions take place within dynamically and spatially controlled cell clusters both within and without lymphoid tissues (Pittet et al, 2018; Bala et al, 2022; Hor & Germain, 2022; Mihlan et al, 2022). In the lymph node, chemokine-driven migration and localization within discrete environments drive T-cell activation and differentiation (Groom et al, 2012; Ugur & Mueller, 2019; Leal et al, 2021). This is most elegantly illustrated in the dynamic choreography of cell migration during the formation of a germinal center (Victora & Nussenzweig, 2022). Although less well understood, similar localization and clustering

of T cells can regulate T-cell activation and impact T-cell fate in peripheral tissues (Natsuaki et al, 2014; Prizant et al, 2021) and tumors (Jansen et al, 2019; Di Pilato et al, 2021).

The development of IVM has been instrumental in revealing the spatial and temporal dynamics of cell migrations that regulate and mediate immune responses (Pittet et al, 2018; Bala et al, 2022; Hor & Germain, 2022; Mihlan et al, 2022). One limitation to IVM is the number of specific cell types that can be simultaneously imaged. This is due in part to the availability of mice expressing cell type–specific fluorescent proteins that can be spectrally distinguished. A second limitation to IVM is the ability to image the same tissue site in a single animal over the course of an entire immune response. As a result, many studies focus on a single time point often at the peak of the response.

In this report, we have addressed these limitations by creating two new mouse strains expressing fluorescent proteins within specific immune cell types. IEbeta-mAmetrine drives the expression of mAmetrine in MHC class II–positive cells, and CD8beta-LSSmOrange drives the expression of LSSmOrange in conventional CD8-positive cells. In addition, we have modified the existing dermal imaging protocol (Overstreet et al, 2013; Prizant et al, 2021) so that we can image the same region of mouse skin dermis over the entire length of an inflammatory response. We applied these techniques to mouse co-expressing markers for MHC class II, CD11c, CXCL10, and CD4 T cells to quantify the formation of CXCL10-positive cell clusters, elaboration of different MHC class II–positive cells within these clusters, accumulation of CD4 T cells within these clusters, and the dissipation of these T-cell activation niches (Prizant et al, 2021; Bala et al, 2022) as the inflammatory response wanes.

## Results and Discussion

### IEbeta-mAmetrine and CD8beta-LSSmOrange mice

To generate new animals for intravital imaging for immune cell types, we first screened candidate fluorescent proteins for two-photon excitation and spectral overlap. The goal was to identify fluorescent proteins that were spectrally distinct from commonly

---

Center for Vaccine Biology and Immunology and Department of Microbiology and Immunology, University of Rochester Medical Center, Rochester, NY, USA

Correspondence: jim_miller@urmc.rochester.edu

The following labels appear within the figure:

**A** Total cells

Lymph Node — WT B6 — IEβ-Ametrine
Spleen — WT B6 — IEβ-Ametrine
Lung — WT B6 — IEβ-Ametrine

Ametrine (y-axis), Class II IA^b (x-axis)

**B** CD19+ B cells

Lymph Node — WT B6 — IEβ-Ametrine
Spleen — WT B6 — IEβ-Ametrine
Lung — WT B6 — IEβ-Ametrine

Ametrine (y-axis), Class II IA^b (x-axis)

**C** CD3-/CD19- cells

Lymph Node — WT B6 — IEβ-Ametrine
Spleen — WT B6 — IEβ-Ametrine
Lung — WT B6 — IEβ-Ametrine

Ametrine (y-axis), Class II IA^b (x-axis)

**D** Ametrine (median FI) vs Endogneous Class II IAb (median FI)

**E** Epidermis
**F**
**G** Thymus
**H**
**I**
**J**

used fluorescent proteins (such as GFP and YFP) so that any animal created could be crossed to common existing strains. We settled on using mAmetrine and LSSmOrange (see Table S1).

MHC class II reporter strains have been previously generated by fusing GFP to the carboxy-terminus of IAbeta and have been useful for tracking the intracellular localization and turnover of class II proteins (Litingtung et al, 2002; Chieppa et al, 2006; Bannard et al, 2016). But such fusion proteins have the potential to induce misregulation of the target protein. To generate a fluorescent marker strain that does not interfere with MHC class II protein structure, we took advantage that C57BL/6 mice maintain a fully functional MHC class II IEbeta gene, but do not express an IE protein because of a defective IEalpha gene. When these mice were reconstituted with a functional IEalpha gene, normal and functional cell surface expression of MHC class II IE was detected (Le Meur et al, 1985; Pinkert et al, 1985; Yamamura et al, 1985). Therefore, we chose to use CRISPR to knock the mAmetrine coding sequence (including a beta globin 3′UTR and polyA addition site) into the start site of IEbeta translation, conferring the genetic locus control of IE to mAmetrine (Fig S1A and Table S2). Although this construct knocks out IEbeta expression, it should have no impact on the expression of class II, as C57BL/6 mice do not normally express IE and should continue to express normal levels of endogenous MHC class II IA proteins.

A reporter strain expressing TdTomato under the control of the CD8alpha gene has been previously generated (Mohan et al, 2017). However, CD8alpha is not a T cell–specific lineage marker and CD8alpha-alpha homodimers can be expressed on a variety of cell types, including dendritic cells, natural killer cells, interstitial epithelial lymphocytes, and gamma delta T cells (Gangadharan & Cheroutre, 2004; Srinivasan et al, 2024). In contrast, CD8beta expression is highly restricted to conventional CD8 T cells. Therefore, we used CRISPR to knock in a cassette containing an IRES and the LSSmOrange coding region into the 3′UTR of the endogenous CD8beta gene (Fig S1B and Table S3). This construct will produce a bicistronic message that should allow for the normal expression of CD8beta from the 5′ Kozak start sequence and LSSmOrange from the IRES in the same mRNA.

### Expression of IEbeta-mAmetrine

To determine whether the expression of mAmetrine was a faithful reporter of MHC class II expression, we harvested cells from lymph nodes, spleen, and lungs from WT C57BL/6 mice and from homozygous IEbeta-mAmetrine mice and stained them for endogenous MHC class II IA[b]. As can be seen in Fig 1A, there is a good concordance between mAmetrine expression driven from the IEbeta knockin and endogenous IA[b]. Within these tissues, the

predominant population of B cells co-express IA[b] and mAmetrine (Fig 1B). To highlight the myeloid class II–positive cells, we gated on CD3[−] and CD19[−] cells. There is more heterogeneity in the levels of class II expression in this population, but again there is a good concordance between mAmetrine and IA[b] (Fig 1C). The expression of mAmetrine did not impact the development of CD4 single-positive cells in the thymus (Fig S2A–C) or the relative number of different class II–positive antigen-presenting populations in the spleen, lymph node, and lung (Fig S3A and B). Taking advantage of the different level of endogenous class II on these subpopulations in different tissues, we found a significant correlation between the level of endogenous IA[b] and mAmetrine in the IEbeta-mAmetrine mice (Fig 1D). The correlation is not perfect likely because of differential half-lives of IA[b] and mAmetrine proteins.

To assess the expression of IEbeta-mAmetrine within intact tissue, we imaged the epidermal layer of mouse ear skin by IVM (Fig 1E and F). The mice were maintained on a C57BL/6-Albino background to eliminate spectral interference with melanin (Sabino et al, 2016). The expression of mAmetrine clearly identified the MHC class II–positive Langerhans cells (LC) that reside above the collagen-rich dermis and display distinct dendritic cell morphology as be seen in the higher magnification image (Fig 1E and F) (Romani et al, 2010; Tong et al, 2015). To determine whether mAmetrine can also be used to identify MHC class II–positive thymic epithelial cells, the thymus was removed from IEbeta-mAmetrine mice and imaged ex vivo on the two-photon microscope. Cells expressing mAmetrine with epithelial cell morphology were clearly visible both in the cortical region of the thymus (Fig 1G and H) and deeper in the medullary region (Fig 1I and J) (Yang et al, 2006). Collectively, the flow data and the tissue imaging data indicate that mAmetrine expression is a faithful reporter of MHC class II expression in IEbeta-mAmetrine mice.

### Expression of CD8beta-LSSmOrange

To determine whether the expression of LSSmOrange was a faithful reporter of CD8beta expression, we harvested cells from spleen and thymus from WT C57BL/6 mice and from heterozygous (±) and homozygous (++) CD8beta-LSSmOrange mice and stained them for endogenous CD8beta. There is a good concordance between LSSmOrange and CD8beta in spleen cells (Fig 2A and B). In the thymus, LSSmOrange expression is retained in single-positive CD4 T cells, presumably because of a longer half-life of the LSSmOrange protein compared with CD8 (Fig 2C). This residual LSSmOrange expression is lost in CD4 T cells in the periphery (Fig 2B). To assess the expression of CD8beta-LSSmOrange within intact tissue, we crossed these mice to IEbeta-mAmetrine mice and imaged intact cervical lymph nodes ex vivo on the two-photon

---

**Figure 1.   Expression of mAmetrine in the IEbeta-mAmetrine mice is a faithful reporter for MHC class II.**
Representative flow cytometry of cells isolated from lymph nodes, spleens, or lungs of WT or homozygous IEbeta-mAmetrine mice and stained for endogenous class II I-A[b] (FITC) and displayed with mAmetrine expression. **(A)** Total cells from each tissue. **(B)** Samples gated on CD19[+] B cells. **(C)** Samples gated on CD3[−], CD19[−] cells. **(D)** Median fluorescence intensity of mAmetrine by endogenous MHC class II IA[b] expression in gated subpopulations shown in Fig S3 (B cells, cDC1, cDC2, CD103 DC, CD11c+ cells, CD11b+/CD11c+ cells from the lymph node, spleen, and lung of IEbeta-mAmetrine mice) (R$^2$ = 0.62, $P$ < 0.0001). **(E, F)** Intravital imaging of ear skin epidermis showing expression of IEbeta-mAmetrine mice in Langerhans cells. Lines in the top panel are hair follicles that autofluoresce in multiple channels. The bottom panel is higher magnification image from a separate site illustrating LC morphology. **(G, H, I, J)** Thymus was isolated from IEbeta-mAmetrine mice and imaged on the two-photon microscope ex vivo. **(G, H, I, J)** corresponds to the cortical region near the surface and (I, J) corresponds to the medullary region deeper in the tissue with low- and high-magnification images shown for both regions. **(E, F, G, H, I, J)** Bars are 100 μm in (E), 50 μm in (G, I), and 20 μm in (F, H, J).

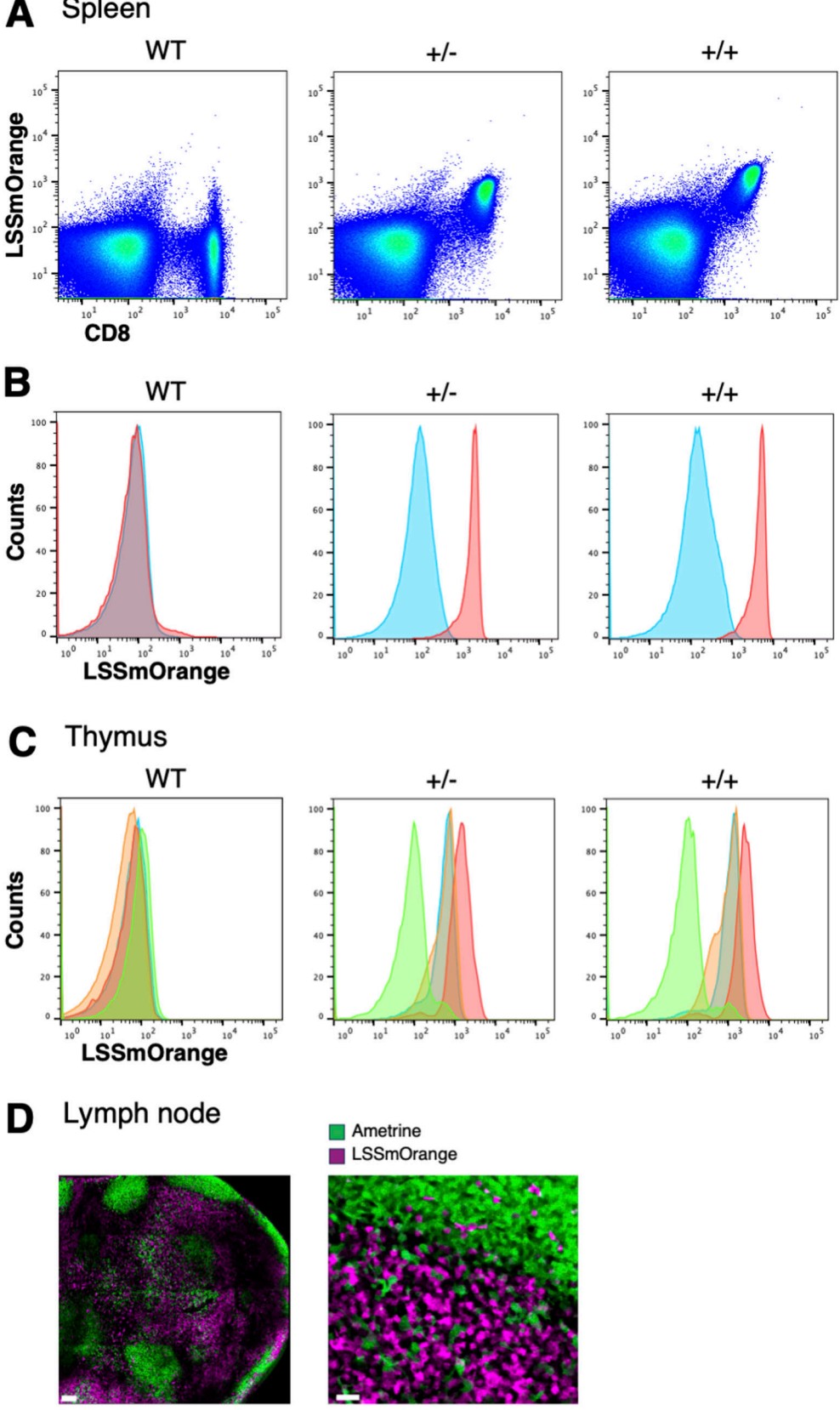

**Figure 2. LSSmOrange is a faithful reporter of CD8 expression in CD8beta-LSSmOrange mice.**
**(A)** Representative flow cytometry of splenocytes isolated from WT mice (left), and from heterozygous (+/−, center) and homozygous (++, right) CD8beta-LSSmOrange mice, stained for CD8beta, and displayed with LSSmOrange expression. **(B)** Histograms of LSSmOrange expression in CD4 (blue) and CD8 (red) cells from spleen of WT, +/−, and ++ mice. **(C)** Histograms of LSSmOrange expression in CD4 single-positive (blue), CD8 single-positive (red), CD4/CD8 double-positive (orange), and CD4/CD8 double-negative (green) cells from thymus of WT, +/−, and ++ mice. **(D)** Cervical lymph node was isolated from mice co-expressing CD8beta-LSSmOrange and IEbeta-mAmetrine and imaged on the two-photon microscope ex vivo. mAmetrine expression (green) is localized primarily in peripheral B-cell follicles, and LSSmOrange (magenta) expression is localized primarily within the central T-cell zone. A close-up of the T:B border is shown in the right panel. Bars are 100 μm in the left panel and 20 μm in the right.

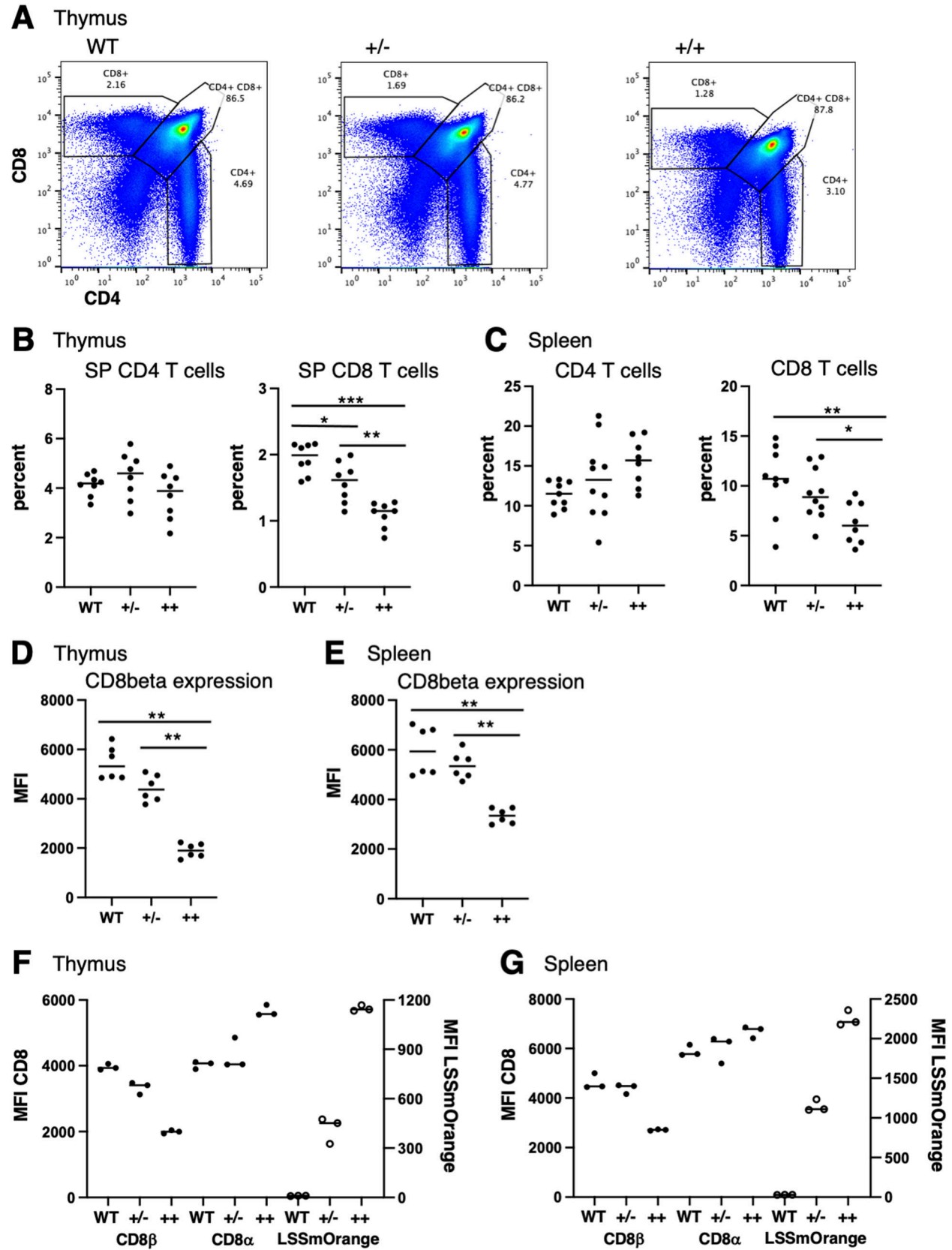

microscope. As can be seen in Fig 2D, mAmetrine expression is localized primarily in peripheral follicles containing class II–positive B cells and LSSmOrange expression is localized primarily within the central T-cell zone. Taken together, these data indicate that the expression of LSSmOrange in CD8beta-LSSmOrange mice accurately reflects the expression of endogenous CD8beta expression.

The homozygous expression of LSSmOrange did result in a selective decrease in single-positive CD8 T cells, starting in the thymus and persisting in the periphery (Fig 3A–C). Along with the decrease in CD8 T cells, there was a corresponding decrease in endogenous CD8beta expression in both single-positive thymocytes and splenic CD8 T cells (Fig 3D and E). This did not appear to be a result of the increased expression of the fluorescent protein reporter, as the level of LSSmOrange was proportionally increased in homozygous mice compared with heterozygous mice (Fig 3F and G). Interestingly, the level of CD8alpha was also increased in homozygous mice (Fig 3F and G). Conventional CD8 T cells express primarily CD8 alpha-beta heterodimers, rather than CD8 alpha-alpha homodimers, suggesting that beta may be normally expressed in excess to drive primarily heterodimer formation (Devine et al, 2000; Srinivasan et al, 2024). Thus, one possible explanation of these observations is that the conformation of the bicistronic CD8-LSSmOrange produced from the knockin construct interferes with translation of CD8, but not LSSmOrange. In heterozygous CD8beta-LSSmOrange mice, there may be little impact on cell surface CD8 expression, as sufficient CD8beta is present to dimerize with CD8alpha. Consistent with this interpretation, there is little impact on CD8 surface expression in hemizygous CD8beta knockout mice (Fung-Leung et al, 1994). However, in CD8beta-LSSmOrange homozygous mice, the level of CD8beta is reduced below this threshold, resulting in a reduction of CD8beta at the cell surface and a slight increase in CD8alpha homodimers. CD8 alpha-beta heterodimers (and not CD8 alpha-alpha homodimers) are required for efficient positive selection in the thymus (Fung-Leung et al, 1994; Gangadharan & Cheroutre, 2004; Cheroutre & Lambolez, 2008), so this loss in CD8beta expression could account for the reduced frequency of CD8 T cells in homozygous CD8beta-LSSmOrange mice.

### The CD8 T cells in CD8beta-LSSmOrange mice are functional

Because LSSmOrange expression can result in a decrease in CD8 cell surface expression and CD8 T-cell numbers particularly in homozygous mice, we tested whether the CD8beta-LSSmOrange mice were immunocompetent. WT mice and homozygous and heterozygous CD8-LSSmOrange mice were infected intranasally with influenza virus. After 10 d, the mice were injected with APC-labeled CD45 to stain cells in the vascular tissue (Anderson et al, 2014) and lungs were harvested and analyzed by flow cytometry to enumerate the recruitment of CD4 and CD8 cells into the lung tissue. Lungs from uninfected mice have very few CD4 or CD8 T cells within lung tissue, and after infection, similar numbers of both CD4 and CD8 T cells were recruited into the lung in WT, homozygous, and heterozygous mice (Fig 4A). In addition, the CD8 T cells that were recruited into infected lung tissue in WT, homozygous, and heterozygous mice expressed similar levels of markers associated with T-cell activation and tissue localization (Fig 4B). These results suggest that although there are fewer CD8 T cells in the homozygous mice (Fig 3A–C), they are capable of generating a robust response to influenza infection.

To further test the responsiveness of CD8 T cells in CD8beta-LSSmOrange mice, we crossed these mice to OTI TCR transgenic mice. Both WT and heterozygous CD8beta-LSSmOrange OTI mice showed the relative increase in CD8 single-positive cells in the thymus, increase in CD8 T cells in the spleen, and selective expression of Vbeta5 and Valpha2 (Fig S4A and B), indicative of efficient positive selection of OTI T cells (Hogquist et al, 1994). To determine whether these OTI T cells were functional, we labeled them with eFluor 450, stimulated them in vitro with antigen and antigen-presenting cells, and analyzed the dilution of eFluor 450 as an indicator of cell division. Both WT and CD8beta-LSSmOrange heterozygous OTI cells generated the same pattern of eFluor dilution, indicating that they proliferated equivalently (Fig S4C). Finally, we co-adoptively transferred naïve WT OTI cells expressing GFP and OTI cells from heterozygous CD8beta-LSSmOrange mice into WT mice and infected the mice with recombinant influenza virus containing the ovalbumin epitope for OTI cells (Reilly et al, 2020). At day 10, after vascular staining with anti-CD45, we harvested spleen, draining lymph node, and lung and determined the ratio of GFP-positive WT to CD8beta-LSSmOrange OTI cells within the lung tissue. Although the ratio of WT to CD8beta-LSSmOrange OTI cells is slightly increased, both WT and CD8beta-LSSmOrange OTI cells expanded within this competitive environment (Fig 4C) and the OT1 T cells that were recruited to lung tissue expressed similar levels of markers associated with T-cell activation and tissue localization (Fig 4D). Collectively, these data indicate that the CD8 T cells in heterozygous CD8beta-LSSmOrange mice are functional.

### Sequential imaging protocol

To follow the kinetics of inflammation within the dermis, we first needed to establish a method to repeatedly image the same region of the mouse ear without inducing additional inflammation. Previous protocols for intravital imaging of the mouse ear used hair removal and/or adhesive tape for immobilization (Li et al, 2012; Overstreet et al, 2013; Pineda et al, 2015; Li et al, 2018; Prizant

---

**Figure 3. Homozygous expression of CD8beta-LSSmOrange results in loss of CD8 T cells and down-regulation of CD8beta expression.**
**(A)** Representative flow cytometry of thymocytes isolated from WT (left), and from heterozygous (+/−, center) and homozygous (++, right) CD8beta-LSSmOrange mice and stained for CD8beta and CD4. Gates and percentages for single-positive CD4, single-positive CD8, and double-positive cells are shown. **(B, C)** Percentage of single-positive CD4 and CD8 thymocytes (B) or of CD4- and CD8-positive splenocytes (C) from individual WT, +/−, and ++ CD8-LSSmOrange mice. **(D, E)** Mean fluorescence intensity (MFI) of CD8beta expression in single-positive CD8 thymocytes (D) or in CD8-positive splenocytes (E) from individual WT, +/−, and ++ CD8-LSSmOrange mice. **(F, G)** MFI of CD8beta and CD8alpha (closed circles, left-hand axis) and MFI of LSSmOrange (open circles, right-hand axis) in single-positive CD8 thymocytes (F) or in CD8-positive splenocytes (G) from individual WT, +/−, and ++ CD8-LSSmOrange mice (***$P < 0.0005$; **$P < 0.005$; *$P < 0.05$).

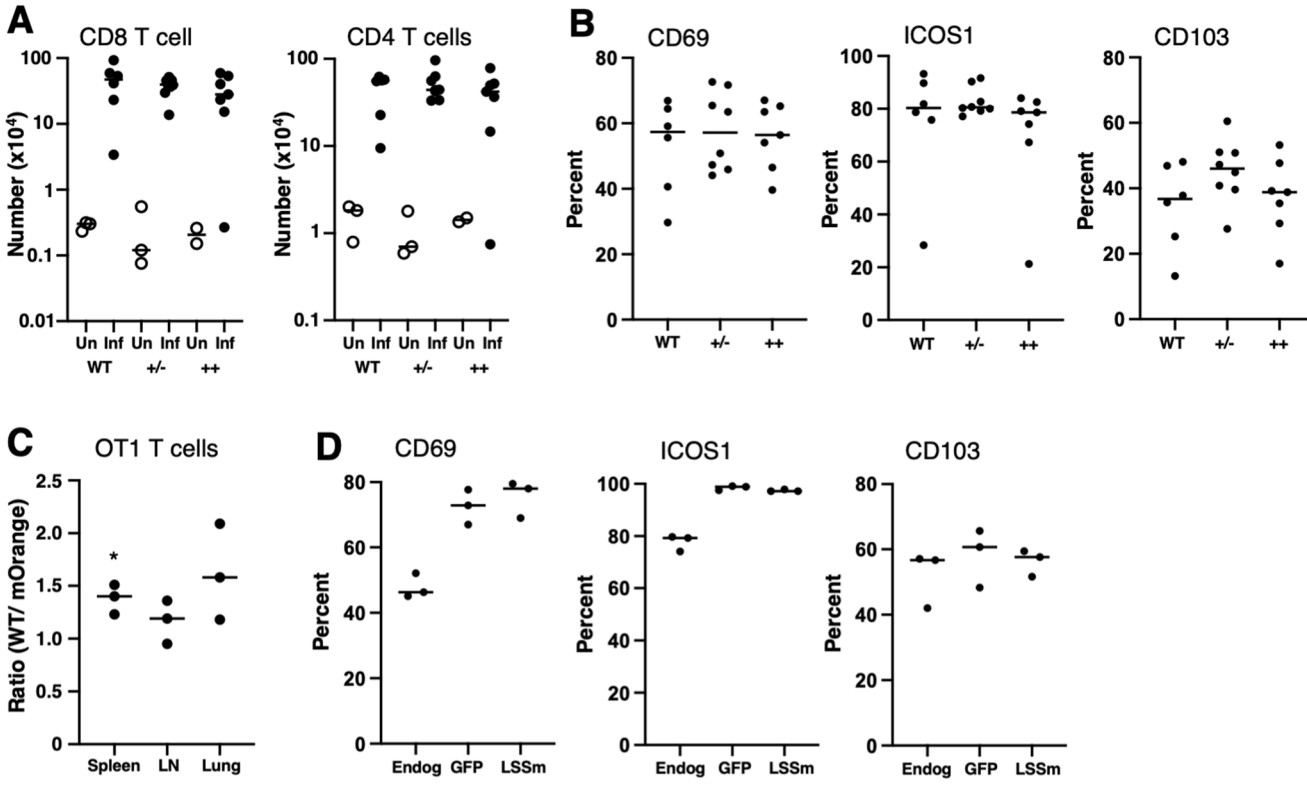

**Figure 4. CD8 T cells from CD8beta-LSSmOrange mice can respond to influenza infection.**
**(A, B)** WT, heterozygous (±), and homozygous CD8beta-LSSmOrange (++) mice were intranasally infected with influenza, and on day 10, cells were isolated from lungs from uninfected (Un, open symbols) and infected (Inf, filled symbols) mice and analyzed by flow cytometry. **(A, B)** Number of lung tissue–resident CD8 T cells (left) and CD4 T cells (right) is shown in (A), and the percentage of lung-resident CD8 T cells expressing activation markers in influenza-infected mice is shown in (B). Data are combined from two separate experiments, n = 2–8. There were no significant differences between WT, +/−, or ++ infected mice. **(C, D)** T cells were isolated from heterozygous CD8beta-LSSmOrange OTI TCR transgenic and from CAG-GFP OTI TCR transgenic mice, and co-adoptively transferred into WT mice, and the recipient mice were infected with recombinant influenza expressing OVA peptide and analyzed by flow cytometry 10 d after infection. **(C, D)** Ratio of WT CAG-GFP OTI to CD8-LSSmOrange OTI T cells in spleen, draining lymph nodes, and lung tissue is shown in (C), and the expression of activation markers on lung-resident WT OTI T cells (GFP), CD8beta-LSSmOrange OTI T cells (LSSm), and endogenous (non-OT1) CD8 cells (Endog) is shown in (D). **(C)** Data are from three individual mice; the ratio in spleen, but not lymph node or lung, is significantly different from the starting ratio of 1.02 (*$P < 0.05$) in (C); and the expression of activation markers in the WT and CD8-LSSm-Orange OT1 cells is not significantly different.

et al, 2021). However, hair removal and repeated tape stripping can disrupt barrier function in the skin and induce inflammation (Nesovic et al, 2024). Therefore, we established a procedure to immobilize the ear using only a foam pad and coverslip (see the Materials and Methods section for details). To confirm that repeated imaging using this procedure did not induce inflammation, mice expressing IEbeta-mAmetrine were crossed to mice expressing CD11c-Venus (Lindquist et al, 2004) and REX3 (CXCL10-BFP/CXCL9-RFP) (Groom et al, 2012) and the same area of skin was repeatedly imaged over the course of 26 d. Using vascular and hair follicle landmarks, we are able to localize and collect intravital microscopic images over the same 850 × 850 μm of the ear dermis over the 26-d time course (Fig S5A). Class II–positive and CD11c-positive dermal dendritic cells are motile (Kissenpfennig et al, 2005; Ng et al, 2008), and the relative positions of mAmetrine (Fig S5B)- and Venus (Fig S5C)-expressing cells within the tissue do change over time. However, there is no apparent increase in the number of MHC class II–positive cells (IEbeta-mAmetrine) or CD11c-positive cells. There are a few BFP-positive cells visible in the

images; however, these isolated cells do not form the clusters seen after complete Freund's adjuvant (CFA)–induced inflammation (Prizant et al, 2021; Bala et al, 2022). Thus, the repeated imaging protocol does not induce changes in the expression of the inflammation-associated markers tested, suggesting that there is minimal tissue response to the repeated imaging protocol.

### Inflammation-induced activation of Langerhans cells

As a first test to use the IEbeta-mAmetrine mice to monitor cellular changes after induction of an inflammatory response, we imaged the epidermal, LC layer of the ear skin before and 2 d after induction of inflammation with CFA. Once again, we were able to localize the same tissue area at each time point (Fig 5A). CFA-induced inflammation resulted in a reduction in the density (Fig 5B and D) of LC within the skin. In addition, there was a change in LC morphology, with a retraction of dendrites and rounding of the cell body, noticeable within the remaining LC cells on day 2 (Fig 5C and E). These data are consistent with LC activation in response to

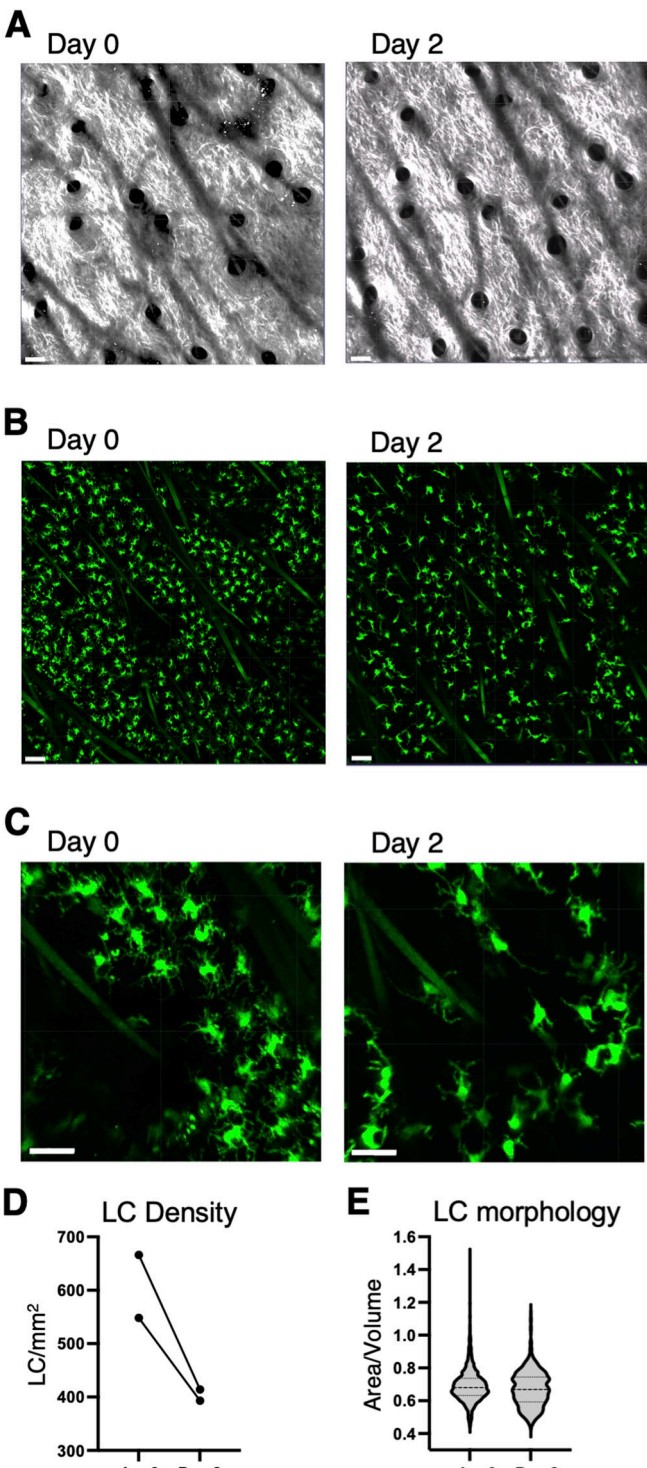

**Figure 5. Activation of Langerhans cells after intradermal immunization with OVA/CFA.**
IEbeta-mAmetrine mice were immunized in the dermis of the ear with OVA/CFA containing a tracer of Alexa 647-OVA. Four hours later, the OVA/CFA depot was localized, an area proximal to the depot and distal to the site of injection was chosen, and an image in the epidermal layer was collected intravitally on the two-photon microscope. Two days later, the same area was identified and reimaged. **(A)** Relative position of hair follicles (dark circles) in the second harmonic generation image confirms imaging of the same location.

innate signals in CFA resulting in changes in motility and ultimate migration from the skin to the draining lymph node (Kissenpfennig et al, 2005; Eidsmo et al, 2009).

### Induction and resolution of peripheral activation clusters

To follow the induction of inflammation in the skin further, we crossed the IEbeta-mAmetrine mice to mice CD11c-Venus and REX3 mice. Although REX3 mice express both CXCL10-BFP and CXCL9-RFP, we have limited our analysis to CXCL10-BFP. Because CXCL10, and not CXCL9, can be induced by type I interferons, CXCL10 is expressed earlier during the inflammatory response. Later, induction of IFN-gamma further induces CXCL10 along with CXCL9 (Groom & Luster, 2011). We and others (Prizant et al, 2021) have found that the subset of CXCL9-RFP–expressing cells that are induced later in inflamed skin primarily co-express CXCL10-BFP. Because we are using this as a marker for the induction and resolution of the chemokine-expressing cell cluster, CXCL10-BFP provides a longer and more consistent marker for the cell cluster. The mice were intradermally immunized with OVA in CFA, and then, using vascular and hair follicle landmarks, the same region of the skin was sequentially imaged from 6 to 33 d (Fig 6A). As has been previously reported, CFA immunization induces clusters of CXCL10-expressing cells, termed T-cell activation niches (Prizant et al, 2021; Bala et al, 2022). During this time course, we were able to observe the expansion and contraction of the chemokine-expressing cell clusters that peak on day 12 (Fig 6B and E). In addition, based on the differential expression of IEbeta-mAmetrine, CD11c-Venus, and CXCL10-BFP, we were able to identify four different populations of class II–expressing cells (Fig 6C, D, and F–H). There are few class II–positive cells in the tissue on day 6, when the CXCL10+ cluster has already begun to form. All four class II–positive populations increase in number, localizing primarily within the clusters, and peak on day 12. However, the two CXCL10-positive populations predominate during these early time points (Fig 6F and G). From day 12 to day 33, the size of the clusters diminishes, suggesting an abatement of the inflammatory response (Fig 6B and E). In contrast, the total number of class II–positive cells continues to increase (Fig 6F). By day 33, most of the class II–positive cells reside outside of the clusters. However, cells expressing all three markers (class II, CD11c, and CXCL10) remain primarily within the clusters (Fig 6H).

### Kinetics of T-cell localization within peripheral activation clusters

To image recruitment of T cells, we adoptively transferred OTII Th1 cells expressing OFP into IEbeta-mAmetrine/CD11c-Venus/REX3 mice and immunized them with OVA/CFA (Figs 7 and S6A–H). As above, we imagined the same area of the skin during the

**(B)** Fluorescence image of IEbeta-mAmetrine expression in Langerhans cells. The green lines are autofluorescent hair follicles that fluoresce in multiple channels. **(C)** Enlarged image of LC. **(A, B, C)** Bars are 100 $\mu m$ in (A, B) and 30 $\mu m$ in (C). **(D)** Quantitation of the density of LC (# LC/mm$^2$) from two separate experiments. **(E)** Ratio of surface area to volume was determined for individual LC on day 0 and on day 2 as an indication of morphology changes upon LC activation ($P < 0.0001$).

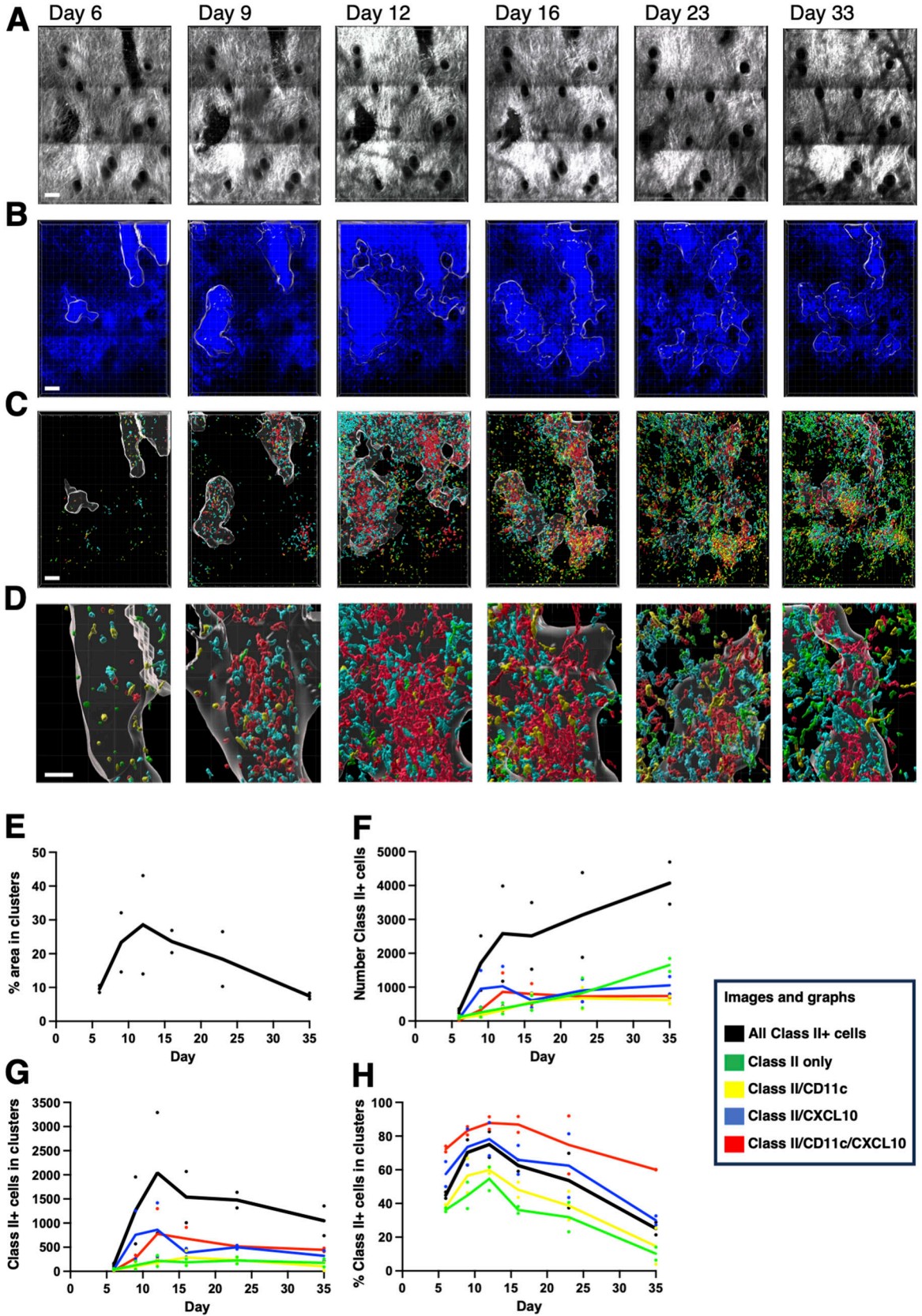

sequential time points (Fig 7A) and observed the formation, expansion, and contraction of the chemokine-expressing cell clusters (Fig 7B and E) and the recruitment of different MHC class II–positive cells into the tissue and localization within the CXCL10 clusters (Fig 7C, D, F, and G). T cells increase dramatically within the tissue from day 3 to day 6 and then slowly wane, with few T cells remaining by about day 30 (Fig 7C, D, and H). The percentage of T cells within the CXCL10 clusters peaks a bit later at about 60% on day 12 (Fig 7C, D, and I). Overall, the kinetics of cluster formation and resolution, the recruitment of class II–positive cells into the tissue, and the accumulation of MHC class II–positive cells into and dispersal from CXCL10 cluster are similar in the presence and absence of exogenous T cells (Fig 8A–C), suggesting that fluorescently labeled T cells are a reasonable representation of the endogenous T-cell response. The addition of the pre-activated Th1 cells does increase the early expression of class II–positive cells within the clusters, possibly because of the increased availability of IFN-gamma (Fig 8B). Notably, the presumptive monocyte dendritic cells, which co-express MHC class II, CD11c, and CXCL10, selectively persist within the clusters in the presence and absence of exogenous T cells (red lines in Fig 8C), even though at this point the overall number of T cells is reduced and the remaining T cells are no longer enriched within the clusters (Fig 7G and H).

In sum, we have developed two new fluorescent protein mouse models that faithfully reflect the expression of MHC class II (IEbeta-mAmetrine) and conventional CD8-positive T cells (CD8beta-LSSmOrange). These strains are now available from the Jackson Laboratory for distribution to the community. We have also established a method for repeated imaging of the same region of the ear dermis and have used the IEbeta-mAmetrine mice crossed to mice expressing CD11c-Venus and CXCL10-BFP and adoptively transferred with OFP labeled to cells to monitor the initiation, amplification, and resolution of an antigen-specific inflammatory response within skin tissue. The use of these mice enabled us to identify several different populations of MHC class II–positive cells; however, there are other populations of immune cells and parenchymal cells present within the tissue that we cannot account for. Sequential imaging of the same site during the course of a T-cell inflammatory response enables us to define the kinetics of this response. Recently, sequential imaging of the hairless region of mouse hind paw has been applied to study the regulation of capillary-associated macrophages (Mesa et al, 2025) and sequential imaging though an abdominal imaging window has allowed the identification of initial events in liver metastasis (Ritsma et al, 2012). The development of additional cell type–specific reporters and sequential IVM imaging techniques will allow for identification of key regulatory events in the initiation and persistence of complicated tissue-restricted diseases.

# Materials and Methods

## Mice

To generate IEbeta-mAmetrine mice, a cassette containing the mAmetrine coding sequence fused to a rabbit beta globin 3′UTR/poly A addition sequence (Fig 1 and Table S2), a sgRNA (5′-TCCTCTCCTGCAGCATGGTG [TGG]-3′), and purified CAS9 were microinjected into fertilized eggs isolated from inbred C57BL/6 mice to knock mAmetrine into the translational start site of the endogenous class II IEbeta gene. Offspring were screened by PCR, and accurate knockin was confirmed by DNA sequencing. The mice were backcrossed to C57BL/6-albino mice (Jackson Laboratory) and maintained in a specific-pathogen-free facility at the University of Rochester Medical Center according to institutional guidelines. These mice are now available from the Jackson Laboratory (Stock No. 039963; IEbeta-mAmetrine).

To generate CD8-LSSmOrange mice, a cassette containing an IRES fused to the LSSmOrange coding sequence (Fig 1 and Table S3), a sgRNA (5′-CTAGCAGGCTATCAGTGTTG [TGG]-3′), and purified CAS9 were microinjected into fertilized eggs isolated from inbred C57BL/6 mice to knock IRES-LSSmOrange into the 3′UTR four nucleotides downstream from the termination codon of the endogenous CD8beta gene. Offspring were screened by PCR, and accurate knockin was confirmed by DNA sequencing. The mice were backcrossed to C57BL/6-albino mice (Jackson Laboratory) and maintained in a specific-pathogen-free facility at the University of Rochester Medical Center according to institutional guidelines. These mice are now available from the Jackson Laboratory (Stock No. 039964; CD8beta-LSSmOrange).

## Tissue preparations and flow cytometry

Single-cell suspensions were prepared from thymus, spleen, and lymph nodes by passing through a 40-$\mu$m nylon mesh filter. Lung cell suspensions were prepared by tissue mincing, by enzymatic digestion in 1 mg/ml type II collagenase (Gibco) and 30 $\mu$g/ml DNase I (Roche) for 45 min at 37°C, and then by passing through a 40-$\mu$m nylon mesh filter. Single-cell suspensions were treated with ACK lysis buffer (0.15 M $NH_4Cl$/1 mM $KHCO_3$/0.1 mM $Na_2$-EDTA,

**Figure 6. Sequential intravital imaging of class II–expressing cells within activation clusters in inflamed ear dermis.**
IEbeta-mAmetrine mice were crossed to CD11c-Venus and REX3 mice and intradermally immunized with ovalbumin in CFA in the ear pinna, and the same area of tissue was sequentially imaged by two-photon microscopy on days 6, 9, 12, 16, 23, and 33. **(A)** Relative position of hair follicles (dark circles) in the second harmonic generation image confirms imaging of the same location. **(B)** Clusters of CXCL10-BFP–expressing cells were identified by nearest neighbor analysis (20 cells within radius of 50 mm) and highlighted with white border. **(C)** Different populations of MHC class II–positive cells were identified based on the differential expression of fluorescent proteins. Green, class II+ only; yellow, class II+ and CD11c+; blue, class II+ and CXCL10+; red, class II, CXCL10+, and CD11c+; white outline, clusters of CXCL10+ cells. **(C, D)** Enlarged images of the CXCL10-BFP+ cluster in the upper right of (C). **(A, B, C, D)** Bars are 100 $\mu$m in (A, B, C) and 20 $\mu$m in (D). Representative data from two independent experiments. **(E, F, G, H)** CXCL10 clusters and class II–positive cells were identified, and the class II+ cells expressing different combinations of CD11c and CXCL10 within and outside the CXCL10 clusters were determined. **(E)** Percentage of imaging area contained within CXCL10 clusters. **(F)** Number of class II+ cells within the entire image. **(G)** Number of class II+ cells within the CXCL10 clusters. **(H)** Percentage of class II+ cells within the clusters. Total number of class II+ cells (black); class II+ only (green); class II+ and CD11c+ (yellow); class II+ and CXCL10+ (blue); and class II, CXCL10+, and CD11c+ (red). Data are average of two separate experiments with individual data points for each time point shown.

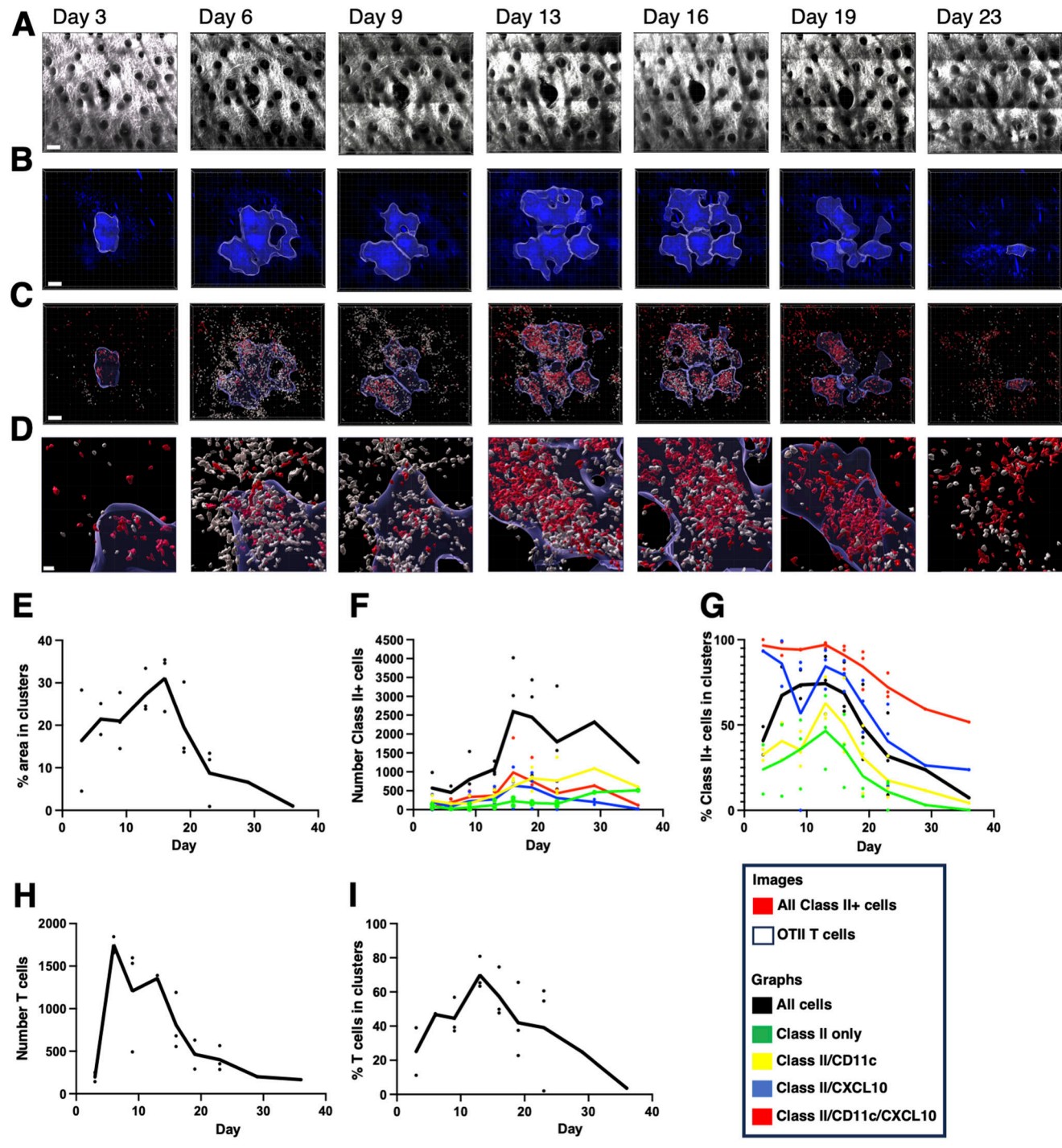

**Figure 7. Sequential intravital imaging of T-cell recruitment to activation clusters in inflamed ear dermis.**
IEbeta-mAmetrine mice were crossed to CD11c-Venus and REX3 mice and were adoptively transferred with 1 × 10⁶ in vitro–activated OTII Th1 cells expressing OFP. The mice were intradermally immunized with ovalbumin in CFA in the ear pinna, and the same area of tissue was sequentially imaged by two-photon microscopy on days 3, 6, 9, 13, 16, 19, and 23. **(A)** Relative position of hair follicles (dark circles) in the second harmonic generation image confirms imaging of the same location. **(B)** Clusters of CXCL10-BFP–expressing cells were identified by nearest neighbor analysis (20 cells within radius of 50 mm) and highlighted with blue border. **(B, C)** All class II–positive cells are shown in red, OTII Th1 cells are shown in white, and the outline of the cluster defined in (B) is shown with a blue outline. **(C, D)** Enlarged images of the data are shown in (C). Representative data from one of three experiments. **(A, B, C, D)** Bars are 100 μm in (A, B, C) and 20 μm (D). **(E, F, G, H, I)** CXCL10 clusters, OTII T cells, and class II–positive cells were identified, and the T cells and the class II+ cells expressing different combinations of CD11c and CXCL10 within and outside the CXCL10 clusters were determined. **(E)** Percentage of image area contained within the CXCL10 clusters. **(F)** Number of class II+ cells within the entire image. **(G)** Percentage of class II+ cells within the clusters. Total number of class II+ cells (black); class II+ only (green); class II+ and CD11c+ (yellow); class II+ and CXCL10+ (blue); and class II, CXCL10+, and CD11c+ (red). **(H)** Number of OTII Th1 cells within the entire image. **(I)** Percentage of T cells within the clusters. Data are the mean of three separate experiments with individual data points for each time point shown. Note there are only two data points for day 3 and day 6 and single data point for days 29 and 36.

# A  % area in clusters

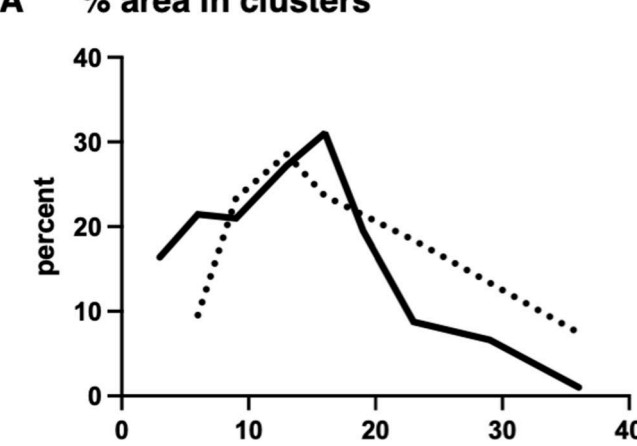

# B  Number Class II+ cells

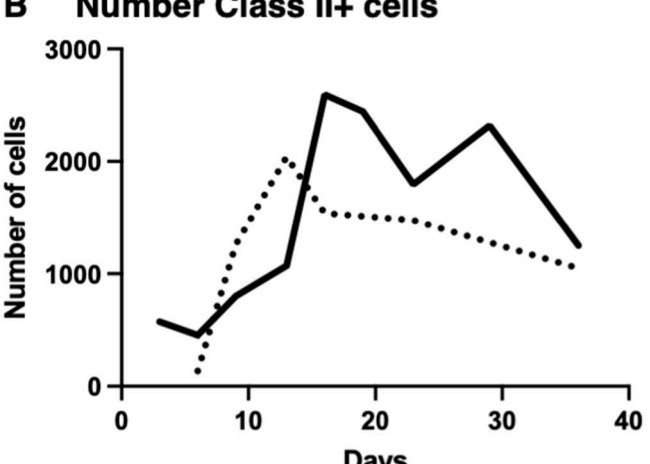

# C  % Class II+ cells in clusters

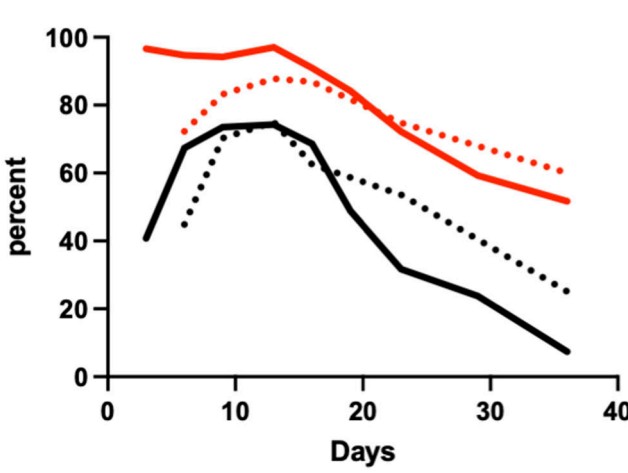

**Figure 8. Comparison of cluster formation in the presence and absence of exogenous T cells.**
Data from Fig 6 (without exogenous T cells, dotted lines) and Fig 7 (with exogenous T cells, solid lines) are overlaid. **(A)** Percentage of image area contained within CXCL10 clusters. **(B)** Number of total class II+ cells with the

pH 7.2) for 5 min to deplete red blood cells. Staining, washes, and resuspensions were performed in 1X PBS with 5% newborn calf serum (HyClone). Cells were prestained with Ghost Dye Violet 510 or Ghost Dye Blue 516 (Tonbo Biosciences) to exclude dead cells and with anti-CD16/CD32 (2.4G2; Tonbo) to block FcR binding. Cells were then stained with fluorescently labeled antibodies as indicated in figure legends. Cells were run on an LSRFortessa or Symphony A1 flow cytometer, and data were analyzed using FlowJo software.

## T-cell assays

CD8beta-LSSmOrange mice were crossed to OTI TCR transgenic mice (Jackson Laboratory), and thymus and spleen were stained with antibodies to CD8, TCR Valpha2, and TCR Vbeta5 to confirm that LSSmOrange-positive CD8 T cells expressing the OTI TCR were appropriately selected. T cells were isolated from spleens and lymph nodes harvested from WT and from heterozygous CD8beta-LSSmOrange mice, were labeled with eFluor 450 (eBioscience), and were stimulated in vitro with 2 µg/ml OVA peptide and irradiated spleen cells. On subsequent days, T-cell division was monitored by dilution of eFluor 450 fluorescence intensity on the flow cytometer.

For influenza T-cell responses, mice were anesthetized with 2,2,2-tribromoethanol (Avertin) via intraperitoneal injection and infected intranasally with 225,000–400,000 egg infectious doses ($EID_{50}$) of influenza A/New Caledonia/20/1999 (H1N1) virus. After 10 d, animals were anesthetized with isoflurane and injected retro-orbitally with 3 µg APC-labeled anti-mouse CD45 (30-F11; Tonbo) and euthanized 3 min later to selectively label vascular cells (Anderson et al, 2014). Cells were isolated from lungs of infected and uninfected mice, the number of interstitial CD8 and CD4 T cells was enumerated, and the expression of activation markers was determined by flow cytometry.

For the competitive adoptive transfer experiment, spleen and lymph node cells were isolated from heterozygous CD8-LSSmOrange or CAG-GFP (Jackson Laboratory) mice co-expressing the OTI TCR transgene. The cells were mixed to yield equal numbers of WT GFP and LSSmOrange CD8 OTI cells, and $1.1 \times 10^6$ total cells containing $2 \times 10^5$ total OTI cells were adoptively transferred by i.p. injection into WT C57BL/6 mice. The mice were then infected with $3 \times 10^3$ $EID_{50}$ HKx31-OVAI expressing the oval-bumin ($OVA^{257–264}$ SIINFEKL) peptide in the stalk of neuraminidase (Reilly et al, 2020) and analyzed 10 d after infection. The ratio of CAG-GFP OTI to CD8-LSSmOrange OTI T cells in spleen, draining lymph nodes, and lungs and the expression of activation markers on lung-resident CD8 T cells were determined by flow cytometry.

## Two-photon microscopic imaging

Images were acquired with an Olympus FV1000-AOM multiphoton system equipped for four-color detection. Fluorescence was collected with an Olympus XLPlanN 25x objective (numerical aperture,

entire image. **(C)** Percentage of total class II+ cells (black lines) and percentage of class II+, CD11c+, CXCL10+ triple-positive cells (red lines) within the clusters.

1.05) and was detected with three proprietary photomultipliers. Fluorescence excitation was achieved by sequential scans with a Spectra-Physics DeepSee-MaiTai HP Ti:Sapphire laser and a Spectra-Physics InSight X3 laser. Laser, dichroic mirror, and band-pass filter configurations are shown in Table S1.

For imaging intact tissue, the thymus was isolated from IEbeta-mAmetrine mice and cervical lymph nodes were isolated from mice co-expressing IEbeta-mAmetrine and CD8beta-LSSmOrange. The tissue was mounted on slides, a z series was collected through the tissue, and single z planes were chosen for display.

For imaging epidermal and dermal layers of mouse ears, mice were anesthetized with isoflurane (induction 4%; maintenance 1.5%, in room air) with an isoflurane vaporizer-ventilation machine (M3000R; Lei Medical). Once the mice were anesthetized, the ear pinna was immobilized between a polyester/acrylic resin foam pad (Home Techpro Vacuum Tech Rug Gripper) and a coverslip without using adhesive or hair removal to avoid any barrier disruption. Body temperature was maintained with a heated water pad (Kent Scientific) and a heating block (WPI). The microscope objective was heated (Bioptechs) to 38°C to maintain constant dermal temperature during imaging (Overstreet et al, 2013).

### Intravital multiphoton imaging

IEbeta-mAmetrine mice were crossed to CD11c-Venus (Lindquist et al, 2004) (Jackson Laboratory) mice and to REX3 (CXCL10-BFP/CXCL9-RFP) mice (Groom et al, 2012), and the triple-positive mice were maintained on a C57BL/6 albino background. The mice were immunized by intradermal injection into ear pinna of 1.76 $\mu$g ovalbumin emulsified in CFA. In some experiments, Alexa 647-ovalbumin (Thermo Fisher Scientific) was added to label the antigen depot. For adoptive transfer experiments, OTII TCR transgenic mice (Jackson Laboratory) were crossed to CAG-mOFP mice (Gossa et al, 2014) and 1–4 × $10^6$ CD4 Th1 cells were adoptively transferred into IEbeta-mAmetrine/CD11c-Venus/REX3 mice before immunization. For in vitro generation of effector Th1 cells, T cells were harvested from lymph nodes and spleens of OTII TCR transgenic mice, and stimulated with 2 $\mu$g/ml ovalbumin peptide and irradiated spleen cells in the presence of IL-2 (10 units/ml; NCI BRB Preclinical Biologics Repository), IL-12 (20 ng/ml; PeproTech), and anti-IL-4 (40 $\mu$g/ml; 11B11; NCI BRB Preclinical Biologics Repository).

For sequential imaging of the same tissue area, the initial localization is based on the organization of the vasculature within a full ear image, and then, by sequential reiteration of vascular and hair follicle orientation, the precise location can be established for subsequent imaging. At each time point, a 1,100 × 1,100 $\mu$m or an 850 × 850 $\mu$m region was stitched together from 3 × 3 or 4 × 4 tiles using Olympus software. Relying on the distribution of collagen fibers detected by second harmonic images and position of hair follicles, images were cropped along the x, y, and z axes to maximize the overlap between time points. Clusters of CXCL10-expressing cells (BFP-positive) were defined in 3D using a Python code that uses the unsupervised machine-learning algorithm DBSCAN (density-based spatial clustering of applications with noise) (Ester et al, 1996). There are two parameters to the algorithm; maximum distance between points, radius (R), and

minimum number of neighbors (N) within R to define as part of the cluster. Density cutoffs of R = 50 $\mu$m and N = 20 neighbors were applied to define the CXCL10 cluster. The images were then spectrally unmixed using Lumos software (McRae et al, 2019) to eliminate bleed over between mAmetrine and BFP or between OFP, Venus, BFP, and RFP, and channels representing MHC class II only (mAmetrine only); class II and CD11c (mAmetrine and Venus), class II and CXCL10 (mAmetrine and BFP); class II and CD11c and CXCL10 (mAmetrine, Venus, and BFP); and when present T cells (mOFP only) were separated. An example of unprocessed images showing individual channels and those after spectral unmixing is shown in Fig S7A–F. Surfaces were generated in 3D software from Imaris (Bitplane) using a seed point diameter setting of 8 $\mu$m to identify individual cells in each channel, and the number of each cell population within and without the CXCL10 clusters was determined. An example of surface generation for individual cell populations is shown in Fig S7G–K.

### Statistics

Statistical tests were done using the Mann–Whitney test in GraphPad Prism.

# Supplementary Information

# Acknowledgements

We thank Drs. Andrea Sant and David Topham for sharing influenza viruses; Drs. Dorian McGavern and Dr. Andrew Luster for providing CAG-OFP and REX3 mice, respectively; Dr Lin Gan for generating the knockin mice; and Dr. Alex Wells for Langerhans cell discussions. We acknowledge the support of the URMC Flow Cytometry Advanced Light Microscopy Cores. This work was supported by Grant # P01 AI102851 from the National Institutes of Health.

### Author Contributions

D Oleksyn: conceptualization, software, formal analysis, validation, investigation, visualization, methodology, and writing—review and editing.
J Miller: conceptualization, formal analysis, supervision, funding acquisition, validation, investigation, visualization, methodology, project administration, and writing—original draft, review, and editing.

### Conflict of Interest Statement

The authors declare that they have no conflict of interest.

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
