## [Reviewer comments · Life Science Alliance]

Intravital imaging the formation and resolution of MHC class II-positive T cell activation niches

David Oleksyn and Jim Miller

DOI: <https://doi.org/10.26508/lsa.202503476>

Corresponding author(s): Jim Miller, University of Rochester

Review Timeline:

Submission Date:	2025-08-04
Editorial Decision:	2025-10-02
Revision Received:	2025-11-21
Editorial Decision:	2025-12-19
Revision Received:	2025-12-22
Accepted:	2025-12-24

Scientific Editor: Tim Fessenden

Transaction Report:

October 2, 2025

Re: Life Science Alliance manuscript #LSA-2025-03476

Dr. Jim Miller
University of Rochester
University of Rochester 601 Elmwood Ave Box 609
Rochester, NY 14642

Dear Dr. Miller,

Thank you for submitting your manuscript entitled "Intravital imaging the formation and resolution of MHC class II-positive T cell activation niches" to Life Science Alliance. The manuscript was assessed by expert reviewers, whose comments are appended to this letter.

As you will see, reviewers agreed that the mouse lines presented here will be valuable resources for researchers interested in intravital imaging of immune cell populations. However each reviewer made important suggestions to strengthen this work. All reviewers requested measurements of effector markers on T cells to corroborate proliferation changes in the influenza model. Reviewers 1 and 2 requested higher magnification images for some figures, and clarification of methodology for cell counting and cluster identification. Reviewers 2 and 3 both remarked that claims on Langerhans cells should be supported with quantitative metrics of cell shape/dynamics, which should be provided or else this claim should be adjusted. We encourage you to increase the n for key assays as requested, if possible. Finally we encourage you to reconsider figure organization according to the suggestions by Reviewer 3. The remaining reviewer comments are left to your discretion.

We hope that the comments below will prove constructive as your work progresses. Thank you for this interesting contribution to Life Science Alliance. We are looking forward to receiving your revised manuscript.

Sincerely,

-- Summary blurb (enter in submission system): A short text summarizing in a single sentence the study (max. 200 characters including spaces). This text is used in conjunction with the titles of papers, hence should be informative and complementary to the title and running title. It should describe the context and significance of the findings for a general readership; it should be

written in the present tense and refer to the work in the third person. Author names should not be mentioned.

B. MANUSCRIPT ORGANIZATION AND FORMATTING:

Reviewer #1 (Comments to the Authors (Required)):

The manuscript "Intravital imaging of the formation and resolution of MHC class II-positive T cell activation niches" reports the generation of two mouse reporter strains (CD8 β -LSSmOrange and IE β -mAmetrine) and their functional validation. Using these novel genetic tools, the authors track over time the formation and resolution of activation clusters in the ear dermis following an inflammatory stimulus.

While the characterization of the mouse models seems mostly compelling, several issues need to be addressed regarding the analysis of the CD8 response to influenza infection, image quality, data quantification, methodology and text editing.

1. Influenza infection: Authors should not only rely on cell numbers but also compare the effector phenotypes and cytotoxic potential of the CD8+ effectors in wt vs CD8 β -LSSmOrange reporter mice.

2. Image magnification:

The results from intravital imaging are difficult to evaluate because most figures are presented at very low magnification. At this scale, fluorescent spots cannot reliably be distinguished as individual cells. Adding higher-magnification insets would significantly improve clarity-for example, in Figure 4 (thymus and lymph node) and in Supplementary Figure S3-similar to what is already shown for Langerhans cells in Figure 4.

The same issue applies to Figures 9, 11, and Supplementary Figure S4. The tiled view of nine fields is informative, but higher-resolution insets of representative regions are needed. Ideally, separate channels showing fluorescence intensity as a pseudocolor density map, would allow readers to better visualize and interpret the cellular composition of the activation clusters.

3. Fluorescence intensity quantification:

The units reported in Figure 8D (#LC/mm²) may not be correct and should be double-checked. Moreover, the fluorescence signal in Figures 9, 11 and Supplementary Figure S4 might be partially saturated to some extent. Under these conditions (low magnification and potential saturation of fluorescent signal), it is difficult to imagine how individual cells could be counted accurately to generate the curves shown in Figures 10, 12-13. Accordingly, if quantification of cell numbers relied on total fluorescence per field of view/fluorescence intensity per cell, the results might be inaccurate. The authors need to provide further methodological details or clarify how they overcame this limitation.

In brief, across multiple figures, the limited resolution and low magnification undermine confidence in the imaging-based analyses. Improving image quality and including detailed methodological explanations would substantially strengthen the manuscript.

4. Methodology

The homeostatic temperature of mouse ears is lower than body temperature (approximately 33 {degree sign}C versus 37 {degree sign}C). Have the authors verified whether heating the microscope objective to 40 {degree sign}C alters the dermal temperature? If so, there could be alteration of the cell migration behavior.

5. Text editing

Revise the text to correct typographical errors, missing words and references.

Reviewer #2 (Comments to the Authors (Required)):

In this study, the authors establish two new reporter mice that enable in vivo visualization of 1) cells expressing MHC-II and 2)

CD8-expressing T cells. The reporter fluorophores were chosen to be optimally compatible with existing reporter mouse models, enabling cross-breeding and generation of multi-reporter mice. Furthermore, the genetic reporter constructs were rationally designed for minimal interference with protein expression and cellular biology, as well as for specificity. The authors present data characterizing the specificity and efficiency of their reporter signals, and show that the MHC-II reporter has no appreciable effect on cell populations. While the CD8 reporter has an effect on CD8 T cells and CD8 antigen expression, these cells remain capable of response to pathogen challenge in vivo.

A secondary aspect of this study was establishment of a protocol for intravital imaging of the same skin sample over time without inducing tissue damage or inflammation. By immobilizing the ear skin with non-invasive, non-irritating methods, and using vasculature and hair follicles as physiological landmarks, the authors demonstrate time course imaging of the same skin areas both at baseline, and in models of inflammatory challenge.

Claims from each figure / main points:

Fig. 1 - ok

Fig. 2 - yes supported

Fig. S1 - yes supported

Fig. 3 - yes supported

Fig. 4 - yes but could use additional data / controls (see comment 2 bullet 2, comment 11)

Fig. 5 - ok

Fig. 6 - yes supported

Fig. 7 - would benefit from more data for completeness (see comment 5, comment 6)

Fig. S2 - ok, but could use quantification (comment 14)

Fig. S3 - no, would benefit from more data/information (see comment 1, comment 9)

Fig. 8 - ok, but could benefit from additional analyses and higher n (comment 4, 7 bullet 2)

Fig. 9 - ok, but additional information and data (unprocessed data, extent of unmixing, identity of CXCL10-BFP+ cells) are merited (see comment 2 bullet 1, comment 3, comment 8)

Fig. 10 - ok, although n is low (see comment 4)

Fig. 11-12 - issues with data representation and/or interpretation (see comment 2 bullet 1, comment 13)

Fig. 13 - see comment on fig. 12

Fig. S4 - see comments on fig. 11-12

Major comments:

1. In Figure S3, the authors state that there is no inflammation induced by their imaging method, but this is not quantified nor thoroughly supported by the data presented.

- Presumably the Day 0 sample is a negative control for inflammation-can the change (or lack thereof) over time be quantified? (timeframe: not time intensive)

- It is not clear whether there are any other cellular infiltrates present, besides those captured by the reporter signals. Can the authors assess inflammation at each time point (or even just the end point) by H&E, or use flow cytometry to survey a diversity of immune cell populations, e.g. neutrophils? (timeframe: 1-2 months)

- Lasers can cause damage and inflammation, and risk of this increases if a tissue is imaged for a longer duration (movies / time) or with higher laser power to image deeper in tissue - is there a laser power / imaging depth / imaging duration limit to this method? This information would help future users understand the applicability and limitations of the method. (timeframe: 1-2 months)

- The current method uses signal from a few reporters, some of which are not apparent in the images shown (is this because there is no inflammation, or because the imaging parameters-BFP, RFP-are not working?). A positive control is merited within the same experiment. Also, BFP and hair follicles are both supposedly shown in blue, but only the follicles are seen.

Designating both BFP and follicles as "blue" could be quite confusing to a reader. Can better deconvolution and/or different coloring be implemented, i.e. using Imaris to process these signals? (timeframe: 1-2 months)

2. In several instances, controls and/or raw imaging data are missing.

- Figures 9, 11, and S4 show highly processed images wherein cells have been binned and masked with surfaces based on their fluorescence traits. As this manuscript deals with new technical tools for imaging, it would be helpful to see data without such heavy processing, where the signals used for cell categorization can be seen in situ. Can the authors show each channel side by side and overlaid, before binning, masking, etc.? (not time intensive)

- Figure 4 demonstrates the new reporter mouse signals in tissue: could the authors include non-fluorescent controls to support the specificity/strength of the new tools? Could they compare homozygous / hemizygous mice (see minor comment 11)? (timeframe: ~1 month)

3. More detailed information about how well the new reporters can be integrated with existing reporters would be of extremely high interest. In Figure 9, the data presented fall short of achieving this end. Showing raw data, and addressing the extent of deconvolution or unmixing needed, would be important. In addition, REX3 mice report on CXCL9-RFP, but this signal is not mentioned in the final analyses presented. Please clarify. (timeframe: <1 month)

4. Statistical power is low throughout the study, with n = 2 mice being used in multiple places. It is hard to draw quantitative

conclusions based on such a low n. (When the text or figure legend says "two experiments," does this mean 2 mice?). (To increase n throughout, timeframe maybe >3 months)

5. Related to comment 4: Would the findings in 7B be statistically significant with a higher experimental n? Since this is a nice way to delineate the impact of the reporter allele on CD8 T cell function, a higher n may be merited to fully characterize the affect, or lack thereof. (timeframe: ~1 month)

6. The authors use cell numbers / tissue infiltrates to support that there is no effect of their new reporter constructs on the T cell response to influenza. It would be good to temper the language here, as all they are measuring is the number of cells accumulating in the tissue. a) Can they state more clearly how this readout reflects function of T cells i.e. an antigen-specific response? b) They do not measure other parameters, like generation of immune memory, cytokine production, cytotoxicity etc., so other more comprehensive measures of "function" have not been thoroughly assessed. (timeframe for additional studies of T cell function: ~1 month, to ~3 months for immune memory readout)

7. Conclusions based on imaging data should employ quantification / statistics to support claims made. Examples: "There is good concordance between LSSmOrange and CD8beta in spleen cells and homozygous mice express higher levels of LSSmOrange (Fig 5A,B)"

"In addition, there was a noticeable change in LC morphology, with a retraction of dendrites and rounding of the cell body (Fig 8C)." (timeframe: not time intensive, unless additional ns are needed)

8. Based on the information in 9C, 11C, we are led to believe that many of the CXCL10-BFP cells are not categorizable as any of the cell types represented in the processed image / masks, especially at the earliest time points. What are these cells? It would be great to complement the imaging data in these figures with flow cytometry data (and unprocessed fluorescence images / overlays), to generate a better idea of the cell types represented, and which cell types can be faithfully imaged with these combined reporter systems. (Also, the CXCL10-BFP clusters' signals appear quite blown out in figures 9, 11, and S4. Can these images be acquired and/or presented with better resolution?) (timeframe: ~2 months)

Minor comments:

9. Is the sequential imaging protocol compatible with models of inflammation that induce significant tissue remodelling? For example, chronic infection can induce changes in the vascular structures within the skin. Could the authors comment on this?

10. The Abstract and Introduction refer to CXCL3 activation clusters, with Figure S4's legend referring to CXCL3-BFP signal; however; everywhere else in the manuscript the BFP signal is noted to be from the CXCL10-BFP reporter in REX3 mice, and CXCL10 clusters are described. Is this a mistake? CXCR3 is the receptor for CXCL9/10, so it seems possible that a mistake was made in the naming the clusters and describing the signals in a few places. Please review for accuracy.

11. In figure 4C, are the mice hemizygous or homozygous for the reporter genes? It would be good to know whether the mice work similarly well for imaging with hemizygous vs. homozygous reporter alleles, as this information would be useful to those using the mice in the future. Perhaps a direct comparison, with equivalent imaging settings, could be presented. This information could be especially important for the CD8 reporter, since this construct at homozygosity seems to have an effect on the endogenous protein / expressing cell. (timeframe: <1 month)

12. "Both WT and CD8beta-LSSmOrange OTI mice showed the same increase in CD8 single positive cells in the thymus cells (supplemental Fig S2A) and increase in numbers of CD8 T cells in the spleen (supplemental Fig S2B) indicative of efficient positive selection of OTI T cells." - in these statements, an increase is noted, but in comparison to what? Clarification, as well as quantification and presentation of all data used for the comparison is merited.

13. The images of T cells (red signal) in figure 11C don't seem to match well with either the quantification in figure 12 or the description in the text. Levels of red cells seem to peak during days 13, 16 and a later peak is supported in S4. Class II+ cells appear to peak around day 6-12. Have the quantifications or categorizations of red / white cells been switched? Please check for accuracy. (If this is rather an issue of how the images are shown, consider splitting the channels and/or not using opaque cell masks, so that signal can be more accurately represented in the images.)

14. Can the authors provide a quantitative comparison of the eFluor 450 dilution / cell proliferation assay, to establish statistical equivalency between the conditions tested?

15. The following statement is made without directly showing data: "This residual LSSmOrange expression is lost in CD4 T cells in the periphery (Fig 5A)." - please show data to specifically support this claim.

16. Minor type-os are present throughout.

17. The figures could be condensed into fewer figures with more panels / figure.

18. Fig. S4 is not referenced in the main text.

19. On Page 6, the CD11c reporter is first referred to as Venus then as YFP. Which is correct?

20. "This decrease in CD8 T cells numbers correlated with a decrease in endogenous CD8beta expression in both single positive thymocytes and splenic CD8 T cells(Fig 6D-E)." - if correlation is not statistically supported, suggest a word such as "corresponded" instead.

Reviewer #3 (Comments to the Authors (Required)):

This manuscript describes the development and use of two new reporter mice of CD8alpha and MHCII using fluorescent proteins that are spectrally distinct from other commonly used in intravital imaging experiments in particular. Intravital imaging of the ear pinna is performed to illustrate specific examples of the value of the new reporter mice.

Overall the reporters described are nicely documented and their use case elaborated on with some informative examples. The figures could be substantially improved with better labeling and grouping the images with their analyses into single figures with informative captions.

In general the introduction lays out the need for the reporters well and provides a clear overview. With regard to methods to return to the same imaging site this has been done in a few contexts that would deserve a mention here: using a "window" (eg. tumour work from van Rheenen lab such as PMID 23115354), along with work in the skin (eg. from Littman lab: PMID 37662387, or Valentina Greco's group PMID 36357619).

I would suggest some reorganization/merging of the figures - there are unnecessarily many of them (13 is excessive) and it reduces clarity in many instances:

Figure 1 can go into supplementary

Figure 2 and 4 to be combined into a figure about the mAmetrine reporter

Figure 3 can go into supplementary

Figure 5 could go into supplementary (C could be labelled in the figure so it is clear that the histograms are in the CD4+ thymocytes) or merged with Figure 6.

Figures 9 and 10 should be merged.

Figures 11 and 12 should be merged.

Some additional points to improve the clarity of the figures.

All figures with imaging data should have larger scale bars and include legends with clear labels of which colour is what population/marker.

- From Figure 2 it is concluded that "the majority of class II positive cells are B cells", but this is not what Fig 2B shows. It is only shown that most B cells are Class II positive. Overall Figure 2 should include gates, specify the fluorophore of the MHCII IAb stain, and could be summarized with some key FACS plots instead.
- In Figure 4 is it not clear from the figure that the lower row is an inset of the top row, and it is not shown which region of the top image the insets correspond to. Please edit accordingly. Please also label what the green and red stains are directly in the figure (recommended to change the colours to avoid red-green combinations).
- Figure 7 cell counts should be put onto a log scale instead of the linear scale shown. It would be important that the authors show, beyond cell counts, that CD8 T cells expressing the reporter are truly function with regard to CD44 upregulation and cytokine production.
- Fig S3. It is not obvious from this figure what landmarks are being used to align the images and how this translates to cell location in B and C. This should be more clearly delineated and cells tracked over time with regard to their location to enable the authors to draw the conclusions they do from these time series data.
- Figure 8. The density units should be shown. Was this done in only a single mouse? The claim is made that dendrites are retracting, but this is not clear from the images shown. If this point is important, this should be quantified. If the purpose of this figure is largely to show that what has been seen before can also be observed with the new reporter, then this could be put into supplementary material or merged with Figure 9.
- Figure 9. The panels showing the clusters with multiple colours are quite hard to follow (cells are very small). How are the white outlines generated?
- Figure 10 - is this quantified from data shown in Figure 9? This was not completely clear. Each data point is from one mouse (ie 2 mice total)? In which case should the sequential points from a single mouse not be joined? Percent of area (in A) is which area - just the total ROI that is being imaged?
- Figure 12. Data points from the same mouse should be joined.

Minor

There are some typos to correct:

"REF" (introduction)

"but do not express and IE protein" (should say "an", second paragraph of results).

"to follow the kinetics of inflammation with the dermis" (should say "within", page 5 start of sequential imaging section).
"Th1 cells does enhances the" (bottom pg 6, the "does" is not needed)

November, 2025

Dear Dr. Fessenden,

We are submitting a revised manuscript #LSA-2025-03476 entitled "Intravital imaging the formation and resolution of MHC class II-positive T cell activation niches" by David Oleksyn and myself for publication in the methods section of Life Science Alliance.

We thank the reviewers for their thoughtful comments and have revised the manuscript accordingly. The major changes that we have made include:

1. We added additional flow cytometry data showing equivalent expression of activation markers on WT and CD8beta-LSSmOrange CD8 T cells following influenza infection (revised Figure 4B and D).

2. We added higher magnification images in revised Figures 1H, 1J, 6D, and 7D and in revised Supplemental Figure S6D and S6H.

3. We added quantitation of Langerhans cell morphology changes to support LC activation following CFA induced inflammation (revised Figure 5E).

4. We elaborated on the methods of imaging site localization, cluster identification, spectral unmixing, and cell identification and quantitation. In addition, we added a new Supplemental Figure (S7) that included an example of image processing, including cluster identification, spectral unmixing, and cell identification.

5. We extensively reorganized the Figures and the manuscript now contains 8 figures and 6 supplemental figures, along with the original 3 supplemental tables.

- Moved Figs 1 and 3 to supplemental figures
- Condensed Figure 2 and parts of Figures 3 and 4 into Figure 1
- Condensed Figure 5 and part of Figure 4 into Figure 2
- Condensed Figure 9 and 10 into Figure 6
- Condensed Figure 11 and 12 into Figure 7

These changes as well as other changes are described in detail in the point-by-point reply to the reviewers' comments.

We have included text files of the revised document and supplemental tables, high resolution TIFF files for the figures and supplemental figures, and a summary blurb.

We hope that you now find these studies suitable for publication in Life Science Alliance.

Sincerely,

Jim Miller
Professor Emeritus

We thank the reviewers for their careful and thoughtful comments on the paper and have revised the manuscript accordingly.

Reviewer 1

The manuscript "Intravital imaging of the formation and resolution of MHC class II-positive T cell activation niches" reports the generation of two mouse reporter strains (CD8 β -LSSmOrange and IE β -mAmetrine) and their functional validation. Using these novel genetic tools, the authors track over time the formation and resolution of activation clusters in the ear dermis following an inflammatory stimulus. While the characterization of the mouse models seems mostly compelling, several issues need to be addressed regarding the analysis of the CD8 response to influenza infection, image quality, data quantification, methodology and text editing.

1. Influenza infection: Authors should not only rely on cell numbers but also compare the effector phenotypes and cytotoxic potential of the CD8⁺ effectors in wt vs CD8 β -LSSmOrange reporter mice.

We added additional flow cytometry data showing equivalent expression of activation markers on WT and CD8beta-LSSmOrange CD8 T cells following influenza infection (revised Figure 4B and D).

2. Image magnification:

The results from intravital imaging are difficult to evaluate because most figures are presented at very low magnification. At this scale, fluorescent spots cannot reliably be distinguished as individual cells. Adding higher-magnification insets would significantly improve clarity-for example, in Figure 4 (thymus and lymph node) and in Supplementary Figure S3-similar to what is already shown for Langerhans cells in Figure 4.

The same issue applies to Figures 9, 11, and Supplementary Figure S4. The tiled view of nine fields is informative, but higher-resolution insets of representative regions are needed. Ideally, separate channels showing fluorescence intensity as a pseudocolor density map, would allow readers to better visualize and interpret the cellular composition of the activation clusters.

We had originally included the wide field view of the images to focus on the overall distribution of cells within the tissue, but the reviewer makes a good point that in these images it is not possible to discern individual cells. We have therefore added panels with higher magnification images of a subregion of the wide field images to revised Figures 1H, 1J, 6D, and 7D and to revised Supplemental Figures S6D and S6H.

3. Fluorescence intensity quantification:

The units reported in Figure 8D (#LC/mm²) may not be correct and should be double-checked. Moreover, the fluorescence signal in Figures 9, 11 and Supplementary Figure S4 might be partially saturated to some extent. Under these conditions (low magnification and potential saturation of fluorescent signal), it is difficult to imagine how individual cells could be counted accurately to generate the curves shown in Figures 10, 12-13. Accordingly, if quantification of cell numbers relied on total fluorescence per field of view/fluorescence intensity per cell, the results might be inaccurate. The authors need to provide further methodological details or clarify how they overcame this limitation. In brief, across multiple figures, the limited resolution and low magnification undermine confidence in the imaging-based analyses. Improving image quality and including detailed methodological explanations would substantially strengthen the manuscript.

We thank the reviewer for catching the arithmetic error in the units for Langerhans cell density and adjusted the y axis in revised Fig 5D accordingly.

We used image analysis software in Imaris to identify and enumerate individual cells within the 3-dimensional images and have clarified this in the methods section. Because the figures are compressed 2-dimensional representations of the 3-dimensional image, there is a greater apparent overlap between cells. The higher magnification images that are now included in the figures, more clearly illustrates the ability to identify individual cells. We also included a new Supplemental Figure (S7) that included an example of image processing, including cluster identification, spectral unmixing, and cell identification.

4. Methodology

The homeostatic temperature of mouse ears is lower than body temperature (approximately 33 {degree sign}C versus 37 {degree sign}C). Have the authors verified whether heating the microscope objective to 40 {degree sign}C alters the dermal temperature? If so, there could be alteration of the cell migration behavior.

We thank the reviewer for catching this error. The temperature of the microscope objective was set at 38°C, which had been previously determined to maintain the ear tissue at approximately 33°C, given the heat distribution through our imaging setup (Overstreet et al (13) Nature Immunology 14:949) and this has been clarified in the text.

5. Text editing

Revise the text to correct typographical errors, missing words and references.

We apologize for the typos that were present in the original manuscript have reviewed the text to identify and correct any typos and missed references.

Reviewer 2

In this study, the authors establish two new reporter mice that enable in vivo visualization of 1) cells expressing MHC-II and 2) CD8-expressing T cells. The reporter fluorophores were chosen to be optimally compatible with existing reporter mouse models, enabling cross-breeding and generation of multi-reporter mice. Furthermore, the genetic reporter constructs were rationally designed for minimal interference with protein expression and cellular biology, as well as for specificity. The authors present data characterizing the specificity and efficiency of their reporter signals, and show that the MHC-II reporter has no appreciable effect on cell populations. While the CD8 reporter has an effect on CD8 T cells and CD8 antigen expression, these cells remain capable of response to pathogen challenge in vivo.

A secondary aspect of this study was establishment of a protocol for intravital imaging of the same skin sample over time without inducing tissue damage or inflammation. By immobilizing the ear skin with non-invasive, non-irritating methods, and using vasculature and hair follicles as physiological landmarks, the authors demonstrate time course imaging of the same skin areas both at baseline, and in models of inflammatory challenge.

Claims from each figure / main points:

Fig. 1 - ok

Fig. 2 - yes supported

Fig. S1 - yes supported

Fig. 3 - yes supported

Fig. 4 - yes but could use additional data / controls (see comment 2 bullet 2, comment 11)

Fig. 5 - ok

Fig. 6 - yes supported

Fig. 7 - would benefit from more data for completeness (see comment 5, comment 6)

Fig. S2 - ok, but could use quantification (comment 14)
Fig. S3 - no, would benefit from more data/information (see comment 1, comment 9)
Fig. 8 - ok, but could benefit from additional analyses and higher n (comment 4, 7 bullet 2)
Fig. 9 - ok, but additional information and data (unprocessed data, extent of unmixing, identity of CXCL10-BFP+ cells) are merited (see comment 2 bullet 1, comment 3, comment 8)
Fig. 10 - ok, although n is low (see comment 4)
Fig. 11-12 - issues with data representation and/or interpretation (see comment 2 bullet 1, comment 13)
Fig. 13 - see comment on fig. 12
Fig. S4 - see comments on fig. 11-12

1. In Figure S3, the authors state that there is no inflammation induced by their imaging method, but this is not quantified nor thoroughly supported by the data presented.

- Presumably the Day 0 sample is a negative control for inflammation-can the change (or lack thereof) over time be quantified? (timeframe: not time intensive)
- It is not clear whether there are any other cellular infiltrates present, besides those captured by the reporter signals. Can the authors assess inflammation at each time point (or even just the end point) by H&E, or use flow cytometry to survey a diversity of immune cell populations, e.g. neutrophils? (timeframe: 1-2 months)
- Lasers can cause damage and inflammation, and risk of this increases if a tissue is imaged for a longer duration (movies / time) or with higher laser power to image deeper in tissue - is there a laser power / imaging depth / imaging duration limit to this method? This information would help future users understand the applicability and limitations of the method. (timeframe: 1-2 months)
- The current method uses signal from a few reporters, some of which are not apparent in the images shown (is this because there is no inflammation, or because the imaging parameters-BFP, RFP-are not working?). A positive control is merited within the same experiment. Also, BFP and hair follicles are both supposedly shown in blue, but only the follicles are seen. Designating both BFP and follicles as "blue" could be quite confusing to a reader. Can better deconvolution and/or different coloring be implemented, i.e. using Imaris to process these signals? (timeframe: 1-2 months)

The reviewer is correct that we cannot rule out that repeated imaging on the same site on the skin did induce some tissue damage and related inflammation. We have acknowledged this caveat and clarified in the text that our analysis is limited to the reporters that we used. That said, these are the reporters that were used in our later experiments to identify class II positive cells within T cell activation niches, so this does serve as a negative control for those experiments.

There are a few BFP-positive cells visible in the images, so we are confident that the BFP channel was active in these experiments. Importantly, these are isolated cells that do not form the clusters seen following CFA induced inflammation.

2. In several instances, controls and/or raw imaging data are missing.

- Figures 9, 11, and S4 show highly processed images wherein cells have been binned and masked with surfaces based on their fluorescence traits. As this manuscript deals with new technical tools for imaging, it would be helpful to see data without such heavy processing, where the signals used for cell categorization can be seen in situ. Can the authors show each channel side by side and overlaid, before binning, masking, etc.? (not time intensive)
- Figure 4 demonstrates the new reporter mouse signals in tissue: could the authors include non-fluorescent controls to support the specificity/strength of the new tools? Could they compare homozygous / hemizygous mice (see minor comment 11)? (timeframe: ~1 month)

We also included a new Supplemental Figure (S7) that included an example of image processing, including cluster identification, spectral unmixing, and cell identification.

For most of the experiments with IEbeta-mAmetrine we used homozygous mice, as, unlike CD8beta-LSSmOrange mice, there was no apparent biological impact on homozygous expression. We have found that both IEbeta-mAmetrine and CD8beta-LSSmOrange expression are sufficiently bright for 2 photon imaging of heterozygous mice. As an example, the lymph node images in revised Figure 2D were taken with heterozygous mice for both IEbeta-mAmetrine and CD8beta-LSSmOrange.

3. More detailed information about how well the new reporters can be integrated with existing reporters would be of extremely high interest. In Figure 9, the data presented fall short of achieving this end. Showing raw data, and addressing the extent of deconvolution or unmixing needed, would be important. In addition, REX3 mice report on CXCL9-RFP, but this signal is not mentioned in the final analyses presented. Please clarify. (timeframe: <1 month)

We have clarified how cell expressing different fluorescent proteins are identified in the methods section. Also, the inclusion of high magnification images in revised Figures 1HJ, 6D, 7D and Supplemental Figures 6D,H better illustrate the identification of individual cells. As mentioned above, we included a new Supplemental Figure (S7) that included an example of image processing, including cluster identification, spectral unmixing, and cell identification.

We apologize for any confusion regarding our reliance on CXCL10-BFP expression rather than CXCL9-RFP expression in REX3 mice. We have clarified this in the text. Briefly, because CXCL10, and not CXCL9, can be induced by type I interferons CXCL10 is expressed earlier during the inflammatory response. Later induction of IFN γ further induces CXCL10 along with CXCL9. We (data not shown) and others (Prizant H et al, 2021) have found that the subset of CXCL9-RFP expressing cells that are induced later in inflamed skin primarily co-express CXCL10-BFP. Because we are using this as a marker for the induction and resolution of the chemokine expressing cell cluster, CXCL10-BFP provides a longer and more consistent marker for the cell cluster.

4. Statistical power is low throughout the study, with n = 2 mice being used in multiple places. It is hard to draw quantitative conclusions based on such a low n. (When the text or figure legend says "two experiments," does this mean 2 mice?). (To increase n throughout, timeframe maybe >3 months)

Some of the experiments were designed as proof of principle, rather than a biological experiment and therefore completed with limited n and no statistical analysis. In the imaging experiments, where the figure legend refers to a given number of experiments, it refers to individual mice.

5. Related to comment 4: Would the findings in 7B be statistically significant with a higher experimental n? Since this is a nice way to delineate the impact of the reporter allele on CD8 T cell function, a higher n may be merited to fully characterize the effect, or lack thereof. (timeframe: ~1 month)

The reviewer is correct that with additional data we might find that the small difference in expansion between WT and CD8beta-LSSmOrange OTI cells after adoptive and influenza infection might become statistically significant. Given the impact of homozygous expression of CD8beta-LSSmOrange expression cell surface CD8 levels and CD8 T cell numbers, it would not be surprising to find that there are some more subtle changes to CD8 T cell repertoire and/or function in heterozygous

mice. But, further evaluation of the mechanism behind these potential defects is beyond the scope of this manuscript.

6. *The authors use cell numbers / tissue infiltrates to support that there is no effect of their new reporter constructs on the T cell response to influenza. It would be good to temper the language here, as all they are measuring is the number of cells accumulating in the tissue. a) Can they state more clearly how this readout reflects function of T cells i.e. an antigen-specific response? b) They do not measure other parameters, like generation of immune memory, cytokine production, cytotoxicity etc., so other more comprehensive measures of "function" have not been thoroughly assessed. (timeframe for additional studies of T cell function: ~1 month, to ~3 months for immune memory readout)*

We added additional flow cytometry data showing equivalent expression of activation markers on WT and CD8beta-LSSmOrange CD8 T cells following influenza infection (revised Figure 4B and D).

7. *Conclusions based on imaging data should employ quantification / statistics to support claims made. Examples:*

"There is good concordance between LSSmOrange and CD8beta in spleen cells and homozygous mice express higher levels of LSSmOrange (Fig 5A,B)"

"In addition, there was a noticeable change in LC morphology, with a retraction of dendrites and rounding of the cell body (Fig 8C)." (timeframe: not time intensive, unless additional ns are needed)

We added quantitation of the changes in Langerhans cell morphology to revised Figure 5E.

8. *Based on the information in 9C, 11C, we are led to believe that many of the CXCL10-BFP cells are not categorizable as any of the cell types represented in the processed image / masks, especially at the earliest time points. What are these cells? It would be great to complement the imaging data in these figures with flow cytometry data (and unprocessed fluorescence images / overlays), to generate a better idea of the cell types represented, and which cell types can be faithfully imaged with these combined reporter systems. (Also, the CXCL10-BFP clusters' signals appear quite blown out in figures 9, 11, and S4. Can these images be acquired and/or presented with better resolution?) (timeframe: ~2 months)*

CXCL10 expression can be induced in a number of both immune and nonimmune cells, and I suspect the early expression of CXCL10 seen in our studies is produced in part by these class II-negative, nonimmune cells. Initial induction of CXCL10 then recruits monocytes and eventually T cells into the tissue. Other labs have analyzed the array of cells that express CXCL10 in different tissue during different inflammatory responses. The goal of our experiments is to validate the use of the IEbeta-mAmetrine animal reporter that we have constructed, so we have focused our analysis on the populations of class II positive cells.

The reviewer is correct to note that the individual cells within images illustrating the CXCL10-BFP clusters are not well resolved. This is in part due to the density of cells within these clusters and because the images shown are a condensed 2D representation of a 3D image. These cells are more easily defined when combined with the other markers used in our study.

Minor comments:

9. *Is the sequential imaging protocol compatible with models of inflammation that induce significant*

tissue remodelling? For example, chronic infection can induce changes in the vascular structures within the skin. Could the authors comment on this?

We added a comment on how the sequential imaging protocol could be extended to other experimental models of tissue responses.

10. The Abstract and Introduction refer to CXCL3 activation clusters, with Figure S4's legend referring to CXCL3-BFP signal; however; everywhere else in the manuscript the BFP signal is noted to be from the CXCL10-BFP reporter in REX3 mice, and CXCL10 clusters are described. Is this a mistake? CXCR3 is the receptor for CXCL9/10, so it seems possible that a mistake was made in the naming the clusters and describing the signals in a few places. Please review for accuracy.

We thank the reviewer for catching this error and the text has been corrected

11. In figure 4C, are the mice hemizygous or homozygous for the reporter genes? It would be good to know whether the mice work similarly well for imaging with hemizygous vs. homozygous reporter alleles, as this information would be useful to those using the mice in the future. Perhaps a direct comparison, with equivalent imaging settings, could be presented. This information could be especially important for the CD8 reporter, since this construct at homozygosity seems to have an effect on the endogenous protein / expressing cell. (timeframe: <1 month)

As discussed above in comment 2, for most of the experiments with IEbeta-mAmetrine we used homozygous mice, as, unlike CD8beta-LSSmOrange mice, there was no apparent biological impact on homozygous expression. We have found that both IEbeta-mAmetrine and CD8beta-LSSmOrange expression are sufficiently bright for 2 photon imaging of heterozygous mice. As an example, the lymph node images in revised Figure 2D were taken with heterozygous mice for both IEbeta-mAmetrine and CD8beta-LSSmOrange.

12. "Both WT and CD8beta-LSSmOrange OTI mice showed the same increase in CD8 single positive cells in the thymus cells (supplemental Fig S2A) and increase in numbers of CD8 T cells in the spleen (supplemental Fig S2B) indicative of efficient positive selection of OTI T cells." - in these statements, an increase is noted, but in comparison to what? Clarification, as well as quantification and presentation of all data used for the comparison is merited.

In most models of TCR transgenic mice, including OTI, there is a selective skewing during thymic development toward single positive cells expressing the appropriate co-receptor for that TCR. For OTI this results in an increase in SP CD8 T cells in the thymus and an increase in CD8 T cells in the periphery. In our experiments it was important to determine whether expression of CD8beta-LSSmOrange impacted CD8 T cell development in OTI TCR transgenics, so it was important to compare WT and CD8beta-LSSmOrange OTI mice, rather than either OTI mouse to nonTCR transgenics. We apologize for the poor wording of the section of the submitted manuscript and have clarified this in the revised text.

13. The images of T cells (red signal) in figure 11C don't seem to match well with either the quantification in figure 12 or the description in the text. Levels of red cells seem to peak during days 13, 16 and a later peak is supported in S4. Class II+ cells appear to peak around day 6-12. Have the quantifications or categorizations of red / white cells been switched? Please check for accuracy. (If this is rather an issue of how the images are shown, consider splitting the channels and/or not using opaque cell masks, so that signal can be more accurately represented in the images.)

We thank the reviewer for catching this error and the figure legend has been corrected. We have also added a color key for the images and graphs in the revised Figure 6 and 7.

14. Can the authors provide a quantitative comparison of the eFluor 450 dilution / cell proliferation assay, to establish statistical equivalency between the conditions tested?

The eFluor experiment in revised Supplemental Figure 4C serves as an in vitro control for the adoptive transfer experiments. The dilution profiles from two separate experiments showing similar proliferation of WT and CD8beta-LSSmOrange OTI cells following in vitro stimulation was sufficient for us to proceed with the adoptive transfer experiment.

15. The following statement is made without directly showing data: "This residual LSSmOrange expression is lost in CD4 T cells in the periphery (Fig 5A)." - please show data to specifically support this claim.

The data that the residual LSSmOrange expression is lost in peripheral CD8 T cells was included in the original version, but not clearly indicated in the text. We have added a reference to the appropriate panel (revised Figure 2B) in the text.

16. Minor type-os are present throughout.

We apologize for the typos that were present in the original manuscript and these have been corrected.

17. The figures could be condensed into fewer figures with more panels / figure.

We have extensively reorganized the Figures and the manuscript now contains 8 figures and 6 supplemental figures, along with the original 3 supplemental tables.

- Moved Figs 1 and 3 to supplemental figures
- Condensed Figure 2 and parts of Figures 3 and 4 into Figure 1
- Condensed Figure 5 and part of Figure 4 into Figure 2
- Condensed Figure 9 and 10 into Figure 6
- Condensed Figure 11 and 12 into Figure 7

18. Fig. S4 is not referenced in the main text.

Reference to supplemental Figure S4 (now SF6) has been included in the main text.

19. On Page 6, the CD11c reporter is first referred to as Venus then as YFP. Which is correct?

We apologize for YFP/Venus confusion in the manuscript. The CD11c mice were generated with the enhanced YFP variant, Venus, and this has been corrected throughout the text.

20. "This decrease in CD8 T cells numbers correlated with a decrease in endogenous CD8beta expression in both single positive thymocytes and splenic CD8 T cells(Fig 6D-E)." - if correlation is not statistically supported, suggest a word such as "corresponded" instead.

We have modified the text to reflect that the loss in CD8 expression corresponds with, rather than correlates with, CD8 T cell number in the text.

Reviewer 3

This manuscript describes the development and use of two new reporter mice of CD8alpha and MHCII using fluorescent proteins that are spectrally distinct from other commonly used in intravital imaging experiments in particular. Intravital imaging of the ear pinna is performed to illustrate specific examples of the value of the new reporter mice.

Overall the reporters described are nicely documented and their use case elaborated on with some informative examples. The figures could be substantially improved with better labeling and grouping the images with their analyses into single figures with informative captions.

In general the introduction lays out the need for the reporters well and provides a clear overview. With regard to methods to return to the same imaging site this has been done in a few contexts that would deserve a mention here: using a "window" (eg. tumour work from van Rheenen lab such as PMID 23115354), along with work in the skin (eg. from Littman lab: PMID 37662387, or Valentina Greco's group PMID 36357619).

We have added the recommended references.

I would suggest some reorganization/merging of the figures - there are unnecessarily many of them (13 is excessive) and it reduces clarity in many instances:

Figure 1 can go into supplementary

Figure 2 and 4 to be combined into a figure about the mAmetrine reporter

Figure 3 can go into supplementary

Figure 5 could go into supplementary (C could be labelled in the figure so it is clear that the histograms are in the CD4+ thymocytes) or merged with Figure 6.

Figures 9 and 10 should be merged.

Figures 11 and 12 should be merged.

We have extensively reorganized the Figures and the manuscript now contains 8 figures and 6 supplemental figures, along with the original 3 supplemental tables.

- Moved Figs 1 and 3 to supplemental figures
- Condensed Figure 2 and parts of Figures 3 and 4 into Figure 1
- Condensed Figure 5 and part of Figure 4 into Figure 2
- Condensed Figure 9 and 10 into Figure 6
- Condensed Figure 11 and 12 into Figure 7

Some additional points to improve the clarity of the figures.

All figures with imaging data should have larger scale bars and include legends with clear labels of which colour is what population/marker.

We added larger scale bars to all the images and included a color key for the images and diagrams in Figures 6 and 7

• From Figure 2 it is concluded that "the majority of class II positive cells are B cells", but this is not what Fig 2B shows. It is only shown that most B cells are Class II positive. Overall Figure 2 should include gates, specify the fluorophore of the MHCII IAb stain, and could be summarized with some key FACS plots instead.

The reviewer is correct in that the original Fig 2B did not show that the majority of class II positive cells are B cells. We have modified the text to state that the predominant population of B cells coexpress class II IAb and mAmetrine.

We included the fluor (FITC) used for the class II staining in the Figure legend

• In Figure 4 is it not clear from the figure that the lower row is an inset of the top row, and it is not shown which region of the top image the insets correspond to. Please edit accordingly. Please also label what the green and red stains are directly in the figure (recommended to change the colours to avoid red-green combinations).

The low and high magnification LC images in revised Fig 1 from different animals, while the low and high magnification images of the thymus (and all other examples in the manuscript) are from the same image. This has been clarified in each figure legend.

We have changed the colors in the lymph node images (now in revised Fig 2D) to green and magenta to eliminate the red-green color combination and have added a color key to the figure.

• Figure 7 cell counts should be put onto a log scale instead of the linear scale shown. It would be important that the authors show, beyond cell counts, that CD8 T cells expressing the reporter are truly function with regard to CD44 upregulation and cytokine production.

We changed the Y axis into a log scale for T cell numbers in revised Figure 4A as recommended.

We added additional flow cytometry data showing equivalent expression of activation markers on WT and CD8beta-LSSmOrange CD8 T cells following influenza infection in revised Figure 4B and D).

• Fig S3. It is not obvious from this figure what landmarks are being used to align the images and how this translates to cell location in B and C. This should be more clearly delineated and cells tracked over time with regard to their location to enable the authors to draw the conclusions they do from these time series data.

For sequential imaging of the same tissue area, the initial localization is based on the organization of the vasculature within a full ear image and then by sequential reiteration of vascular and hair follicle orientation, the precise location can be established for subsequent imaging. We then rely the distribution of collagen fibers detected by second harmonic images and position of hair follicles within the tiled images, to crop along the x, y and z axes to maximize overlap between time points.

We cannot track individual cells within the time points of our kinetic experiments and did not intend to use these data to analyze the migratory capacity of the CD11c+/class II+ cells that were observed in these images. Rather, we were relying on previous studies, referenced in the text, that CD11c+/class II+ dermal dendritic cells are motile to account for the observed changes in distribution of these cells during the time course of our experiment.

• Figure 8. The density units should be shown. Was this done in only a single mouse? The claim is made that dendrites are retracting, but this is not clear from the images shown. If this point is important, this should be quantified. If the purpose of this figure is largely to show that what has been seen before can also be observed with the new reporter, then this could be put into supplementary material or merged with Figure 9.

We added density units of the Langerhans cells before and after exposure to CFA to revised Figure 5D. The Day 0 and Day 2 data are taken from 2 separate mice. We also quantified the changes in the morphology of the Langerhans cells that are remaining in the tissue on day 2 and included that in revised Figure 5E. Although the induction of Langerhans cells activation and migration is well established (see references in manuscript), we have left this experiment as a separate figure as we feel it is a good example of the utilization of the IEbeta-mAmetrine mice.

• *Figure 9. The panels showing the clusters with multiple colours are quite hard to follow (cells are very small). How are the white outlines generated?*

We had originally included the wide field view of the images to focus on the overall distribution of cells within the tissue, but the reviewer makes a good point that in these images it is not possible to discern individual cells. We have therefore added panels with higher magnification images of a subregion of the wide field images to revised Figures 1H, 1J, 6D, and 7D and to revised Supplemental Figures S6D and S6H, where individual cells can be identified.

The white outlines in revised Figure 6 (Blue in revised Figure 7) represent the outline of the BFP cluster. They are generated in Imaris by defining the BFP cluster as a single object and then displaying the 3D outline of that object.

• *Figure 10 - is this quantified from data shown in Figure 9? This was not completely clear. Each data point is from one mouse (ie 2 mice total)? In which case should the sequential points from a single mouse not be joined? Percent of area (in A) is which area - just the total ROI that is being imaged?*
• *Figure 12. Data points from the same mouse should be joined.*

The data in Fig 10 (now revised Figure 6E-H) is quantified from 2 independent experiments, one shown in Figure 9 (now revised Figure 6A-D) and one not shown, For the experiments where we included T cells (revised Figure 7), the data is quantified from 3 independent experiments, one shown in revised Figure 7 and 2 shown in revised Supplemental Figure S6).

The reviewer is correct that the “percent of area” is the area of the cluster compared to the total image area. This has been clarified in the figure legends.

Although we understand the reviewer’s point that the time point data from individual mice could be connected in revised Figures 6 and 7. However, we have chosen not to make this change because it would make the figure too cluttered. For example, Figure 7G currently contains 5 lines representing the average percentage of 5 different class II+ populations. If we included lines for each of the 3 individual mice, the figure would contain 15 different lines.

Minor

There are some typos to correct:

"REF" (introduction)

"but do not express and IE protein" (should say "an", second paragraph of results).

"to follow the kinetics of inflammation with the dermis" (should say "within", page 5 start of sequential imaging section).

"Th1 cells does enhances the" (bottom pg 6, the "does" is not needed)

We apologize for the typos that were present in the original manuscript and these have been corrected.

December 19, 2025

RE: Life Science Alliance Manuscript #LSA-2025-03476R

Dr. Jim Miller
University of Rochester 601 Elmwood Ave Box 609
Rochester, NY 14642

Dear Dr. Miller,

Thank you for submitting your revised manuscript entitled "Intravital imaging the formation and resolution of MHC class II-positive T cell activation niches". As you will see, all reviewers are satisfied with no major requests. We invite you to consider the minor issues noted by Reviewer 2 and address these in the manner of your choosing. We would be happy to publish your paper in Life Science Alliance pending those changes and final revisions necessary to meet our formatting guidelines.

- Please be sure that the authorship listing and order is correct.
- Please add the X and Bluesky handles of your host institute/organization, as well as your own and/or one of the authors, in our system.
- Please mark the corresponding author on the manuscript title page.
- Please be sure that the authorship listing and order are correct.
- Please label clearly the "Conflict of Interest" statement.
- LSA does not permit citation of "data not shown," "manuscript in preparation," "manuscript submitted," etc., in any section of the manuscript. Please remove this phrase at line 275 or include the relevant data.
- Please include either the python code used for cluster analysis, a citation describing this code, or a link to a repository, in the methods section.
- Please add callouts for Figures 8A; S1A-B; S2A-B; S3A-B; S6A-H and S7A-K to your main manuscript text.
- Please correct the typo "egg" at line 382.

LSA now encourages authors to provide a 30-60 second video where the study is briefly explained. We will use these videos on social media to promote the published paper and the presenting author (for examples, see <https://docs.google.com/document/d/1-UWCfbE4pGcDdcgzcmiuJl2XMBJnxKYeqRvLLrLS08s/edit?usp=sharing>). Corresponding or first-authors are welcome to submit the video. Please submit only one video per manuscript. The video can be emailed to contact@life-science-alliance.org

A. FINAL FILES:

B. MANUSCRIPT ORGANIZATION AND FORMATTING:

Thank you for your attention to these final processing requirements. Please revise and format the manuscript and upload materials as soon as you are able.

Sincerely,

Reviewer #1 (Comments to the Authors (Required)):

All concerns raised in my original review have been satisfactorily addressed by the authors.

Reviewer #2 (Comments to the Authors (Required)):

In this study, the authors establish two new reporter mice that enable in vivo visualization of 1) cells expressing MHC-II and 2) CD8-expressing T cells. The reporter fluorophores were chosen to be optimally compatible with existing reporter mouse models, enabling cross-breeding and generation of multi-reporter mice. Furthermore, the genetic reporter constructs were rationally designed for minimal interference with protein expression and cellular biology, as well as for specificity. The authors present data characterizing the specificity and efficiency of their reporter signals, and show that the MHC-II reporter has no appreciable effect on cell populations. While the CD8 reporter has an effect on CD8 T cells and CD8 antigen expression, these cells remain capable of response to pathogen challenge in vivo.

A second aspect of this study was establishment of a protocol for intravital imaging of the same skin sample over time without inducing tissue damage or inflammation. By immobilizing the ear skin with non-invasive, non-irritating methods, and using vasculature and hair follicles as physiological landmarks, the authors demonstrate time course imaging of the same skin areas in models of inflammatory challenge for time courses of multiple weeks.

The study as resubmitted has been improved--the restructuring and condensing of the figures has made a major difference in readability, and the text more accurately reflects the methods and intentions of the experiments presented. Overall this study serves as a nice presentation of two new imaging reporters and demonstrates reasonably well the serial imaging approach. Both of these elements should be of use to the intravital imaging community, and the study should be published.

Minor comments:

- Figure S7 denotes the CD11c reporter as CD11c-YFP (rather than Venus)-please update for consistency with the rest of the manuscript
- In the legend for figure S6, the authors refer to CXCL3-BFP. Should this be CXCL10-BFP?
- The limited nature of the evidence supporting a lack of inflammation induced by the sequential imaging protocol remains a

weakness (Figure S5, previously S3). The analysis doesn't capture overall inflammation and misses key responders to skin challenges (e.g. neutrophils). It is appreciated that the limitations of the analysis are acknowledged in the text, but a more comprehensive view of inflammation would still be valuable, since a lack of "additional inflammation" in general is noted in the text, and the serial imaging protocol is framed as a key advance in this paper.

Reviewer #3 (Comments to the Authors (Required)):

I am satisfied with the authors responses to reviewers.

December 24, 2025

RE: Life Science Alliance Manuscript #LSA-2025-03476RR

Dr. Jim Miller
University of Rochester
University of Rochester 601 Elmwood Ave Box 609
Rochester, NY 14642

Dear Dr. Miller,

Thank you for submitting your Methods entitled "Intravital imaging the formation and resolution of MHC class II-positive T cell activation niches". Our minor formatting requests have been resolved, and we appreciate the clarification on viral titer measures as "egg infectious" (new to this editor). We also note the presence of the DBSCAN code in the original paper cited, and appreciate the clarification here. While it seems somewhat unlikely a reader would use the code verbatim as provided in that paper, this code is indeed widely available. It is a pleasure to let you know that your manuscript is now accepted for publication in Life Science Alliance. Congratulations on this interesting work.

DISTRIBUTION OF MATERIALS:

Again, congratulations on a very nice paper. I hope you found the review process to be constructive and are pleased with how the manuscript was handled editorially. We look forward to future exciting submissions from your lab.

Sincerely,
